# Gut microbiota facilitate chronic spontaneous urticaria

Lei Zhu[1,2,3], Xingxing Jian[4], Bingjing Zhou[1,2,3], Runqiu Liu[5], Melba Muñoz[6,7], Wan Sun [8], Lu Xie [4], Xiang Chen [1,2,3,9], Cong Peng [1,2,3,9], Marcus Maurer [6,7,9] ✉ & Jie Li [1,2,3,9] ✉

Chronic spontaneous urticaria (CSU) comes with gut dysbiosis, but its relevance remains elusive. Here we use metagenomics sequencing and short-chain fatty acids metabolomics and assess the effects of human CSU fecal microbial transplantation, *Klebsiella pneumoniae*, *Roseburia hominis*, and metabolites in vivo. CSU gut microbiota displays low diversity and short-chain fatty acids production, but high gut *Klebsiella pneumoniae* levels, negatively correlates with blood short-chain fatty acids levels and links to high disease activity. Blood lipopolysaccharide levels are elevated, link to rapid disease relapse, and high gut levels of conditional pathogenic bacteria. CSU microbiome transfer and *Klebsiella pneumoniae* transplantation facilitate IgE-mediated mast cell(MC)-driven skin inflammatory responses and increase intestinal permeability and blood lipopolysaccharide accumulation in recipient mice. Transplantation of *Roseburia hominis* and caproate administration protect recipient mice from MC-driven skin inflammation. Here, we show gut microbiome alterations, in CSU, may reduce short-chain fatty acids and increase lipopolysaccharide levels, respectively, and facilitate MC-driven skin inflammation.

Chronic urticaria (CU) is a common and debilitating inflammatory skin disease characterized by recurrent wheals and angioedema[1]. In a recent systematic review, the point prevalence of CU was 0.7%, with considerable regional variations and higher rates in Latin America and Asia[2]. A very recent nationwide cross-sectional study estimated the point prevalence of CU in China at 2.6%[2,3].

Chronic spontaneous urticaria (CSU) is the most common type of CU. CSU is of long duration, 1–4 years on average, with 83%, 55%, and 27% of patients affected for more than 1, 5, and 20 years, respectively[1,4]. The rate of CSU relapse, i.e. the reappearance of CSU signs and symptoms in patients with complete remission after cessation of controller therapy[5], is about 13%[6].

The burden of CSU for patients and society is substantial. Most patients with CSU experience marked quality of life impairment, with detrimental effects on sleep, performance at school and work, sexual health, and emotional and physical well-being[4,7]. Up to half of patients show insufficient response to treatment with an antihistamine or omalizumab, the recommended first and second line therapy for CSU, respectively. Better treatments are needed, for which a better understanding of the pathogenesis of CSU is crucial[8,9].

[1]Department of Dermatology, Xiangya Hospital, Central South University, Changsha, Hunan, China. [2]Hunan Key Laboratory of Skin Cancer and Psoriasis, Furong Labratory, Changsha, Hunan, China. [3]National Clinical Research Center for Geriatric Disorders, Xiangya Hospital, Central South University, Changsha, Hunan, China. [4]Bioinformatics Center, Xiangya Hospital, Central South University, Changsha, Hunan, China. [5]Department of Dermatology, the First people's Hospital of Yancheng, Yancheng Clinical College of Xuzhou Medical University, Yancheng, Jiangsu, China. [6]Institute of Allergology, Charité-Universitätsmedizin Berlin, corporate member of Freie Universität Berlin and Humboldt-Universität zu Berlin, Berlin, Germany. [7]Fraunhofer Institute for Translational Medicine and Pharmacology ITMP, Allergology and Immunology, Berlin, Germany. [8]BGI, Complex building, Beishan Industrial Zone, Yantian District, Shenzhen, China. [9]These authors contributed equally: Xiang Chen, Cong Peng, Marcus Maurer, Jie Li. ✉e-mail: marcus.maurer@charite.de; xylijie@csu.edu.cn

The pathogenesis of CSU has not been fully elucidated. The development of wheals and angioedema is driven by skin mast cells (MCs) and their activation, degranulation and release of mediators, and IgE and IgG autoantibodies are held to activate skin MCs in subpopulations of CSU[10,11]. Skin MCs, in CSU, exhibit increased susceptibility to activation[12,13]. Whether this is due to signals that reduce their activation threshold or reduced exposure to inhibitory signals is currently unknown.

Signals that reduce the activation threshold of MCs include proinflammatory cytokines such as IL33[14] as well as bacterial products like lipopolysaccharide (LPS)[15]. In contrast, the engagement of inhibitory receptors[16], e.g. Siglec6 and 8[17,18], and short chain fatty acids (SCFAs) such as propionate and butyrate inhibit MC activation[19]. LPS, SCFAs, and other factors that influence MC activation thresholds are produced by microbiota of the gut[20].

The gut microbiome maintains intestinal homeostasis, plays a key role in securing the integrity of the intestinal wall, and orchestrates immune balance[21] The microbial homeostasis of the gut depends, in part, on SCFAs produced and released by *Firmicutes* including *Clostridium leptum* and *Roseburia spp*[22,23]. SCFAs regulate the growth and virulence of enteric pathogens, for example *Escherichia coli* and *Klebsiella*[24]. SCFAs also promote epithelial barrier function, by supporting epithelial cell proliferation, stabilizing hypoxia-inducible factor, upregulating the expression of tight junction molecules (e.g. occludin-1, zonula occludens-1[ZO-1], mucin 2 [MUC-2])[25], and producing antimicrobial peptides that protect intestinal epithelial cells[23], thereby preventing its destruction by pathogenic bacteria and harmful substances such as LPS from entering the blood[25,26]. LPS, a component of the cell wall of gram-negative bacteria, facilitates IgE-induced degranulation of MCs via Toll-like receptor 4 (TLR4)[27]. In addition, LPS can directly activate MCs through TLR4 and induce the release of inflammatory mediators such as TNFα and IL6[28].

Importantly, SCFAs also have anti-inflammatory effects. They induce prostaglandin $E_2$ release and expression of the anti-inflammatory cytokine IL10 and inhibit MC activation after IgE- and non-IgE-mediated stimulation via epigenetic regulation, effects that have been proposed to be of benefit in allergic and other MC-mediated diseases[19,29].

That the gut microbiome can contribute to the pathogenesis of CSU is supported by several independent lines of evidence[30–34]. First, recent studies demonstrated that the composition of gut microbiota is altered in CSU, showing decreased bacterial diversity, a known risk factor for allergic disease[35]. Second, serum metabolome analyses in CSU demonstrated reduced levels of SCFAs, known inhibitors of MC activation[35]. Third, gut microbiome changes in CSU patients were correlated with levels of inflammation, disease duration, and response to treatment[36,37]. Fourth, patients with CSU can benefit from probiotic treatment[38]. As of now, the precise role and relevance of the gut microbiome in CSU and the mechanisms involved remain ill characterized and understood.

To address this unmet need, we investigated the gut microbiome of patients with CSU and healthy controls by metagenomics sequencing and targeted metabolomics and assessed the differences for functional relevance. First, we tested their correlation with each other as well as with blood SCFA and LPS levels and clinical features. Next, we evaluated the correlation between gut microbiome alterations, blood levels of LPS, and disease relapse. Then, we adoptively transferred the microbiome of CSU patients, by fecal microbial transplantation(FMT), to mice and assessed them for changes in intestinal permeability, blood LPS levels, and passive cutaneous anaphylaxis(PCA), an IgE-mediated MC-dependent skin inflammatory reaction. Finally, we adoptively transferred mice with *Roseburia hominis* or SCFAs or with *Klebsiella pneumoniae* or LPS and then induced and assessed passive cutaneous anaphylaxis, to identify underlying mechanisms of CSU microbiome effects on MC-dependent skin inflammation in vivo.

## Results
### The microbiome of CSU patients is characterized by reduced diversity and markedly lower levels of SCFA-producing gut bacteria
Fresh stool samples from 26 CSU patients and 26 age- and sex-matched healthy controls were collected for metagenomic sequencing. Plasma samples from 29 CSU patients and 38 healthy controls were collected for metabolic sequencing. Plasma samples from 16 CSU patients and 16 age- and sex-matched healthy subjects were collected for LPS detection (Supplementary Fig. 1, Supplementary Table 1). In CSU patients, the composition of gut microbiota was markedly different as compared to healthy control subjects (Fig. 1a left panel). First, the α-diversity of gut microbiota, i.e. Shannon and Simpson indices, was significantly decreased ($P < 0.05$, Fig. 1a right panel), as assessed by metagenomic sequencing of fecal samples ($n = 26$ each, Supplementary Table 2). Second, CSU patients, as compared to HCs, showed markedly lower levels of major SCFA producers such as *Rikenellaceae*, at the family level, *Alistipes*, at the genus level (Supplementary Fig. 2a, b), and *Roseburia hominis*, at the species level (Fig. 1b).

A differential comparison of 15 core species present in at least 50% of all subjects showed decreased abundance of 10 in CSU, all of which were SCFA-producing bacteria (Fig. 1b, Supplementary Table 2). CSU patients, overall, had significantly fewer SCFA-producing gut microbiota than HCs (Fig. 1c) and significantly lower blood levels of SCFAs including acetate, propanoate, and caproate (Fig. 1d). In CSU, low blood levels of caproate and low levels of SCFA-producing bacteria were linked to increased levels of opportunistic pathogens such as *Klebsiella pneumonia* and *Escherichia coli* (Fig. 1e, Supplementary Fig. 2c, Supplementary Table 2).

### The microbiome of CSU patients shows high levels of opportunistic pathogenic gut bacteria
All major hub species in HCs were SCFA producers (e.g., *Dorea longicatena*, *Lachnospiraceae bacterium*, *Ruminococcus obeum*, and *Ruminococcus sp*., Supplementary Fig. 2d). In contrast, all of the key hub species in CSU were opportunistic pathogens (e.g., *Klebsiella pneumoniae*, *Streptococcus salivarius*, *Streptococcus parasanguinis*, and *Veillenella parvula* (Supplementary Fig. 2d, e). In CSU patients versus HCs, gut levels of conditionally pathogenic bacteria such as *Enterobacteriaceae*, *Peptostreptococcaceae* (Supplementary Fig. 2a, f) were markedly increased. At the genus and species level, CSU patients exhibited increased relative abundance of *Klebsiella* (Supplementary Fig. 2b) and *Klebsiella pneumoniae*, *Bacteroides stercoris* and *Escherichia coli* (Fig. 1b), respectively. Moreover, gut levels of *Klebsiella pneumoniae*, by ROC analyses, distinguished CSU and HCs diagnostically (AUC > 0.7, Supplementary Fig. 2g) and correlated with CSU disease activity (Fig. 1f).

Specifically, CSU patients had more opportunistic pathogens that harm intestinal wall integrity, e.g., *Klebsiella pneumoniae*, *Escherichia coli*, and *Ruminococcus gnavus*, major producers of LPS ($P = 0.002$, Fig. 1g). Differential analyses identified 6 markedly upregulated metabolic pathways in CSU, most prominently the super-pathway of LPS biosynthesis (Fig. 1h left panel). Blood LPS levels in CSU patients versus HCs were higher ($P < 0.05$, Fig. 1h right panel), negatively correlated with SCFA producing bacteria ($P < 0.05$, Fig. 1i left panel), and positively correlated with opportunistic pathogenic bacteria ($P < 0.05$, Fig. 1i right panel). At the species level, blood LPS concentrations were negatively correlated with *Ruminococcus obeum* and *Roseburia hominis*, major SCFA producers ($P < 0.05$, Fig. 1f).

Of 22 CSU patients who had undergone metagenomic sequencing and were followed up for remission and relapse, 17 showed spontaneous remission during the recent five years, and 7 of them experienced relapse, which was linked to low levels of SCFA-producing and high levels of opportunistic pathogenic bacteria ($P = 0.024$, $P = 0.047$, respectively, Supplementary Fig. 2h) and high LPS blood levels

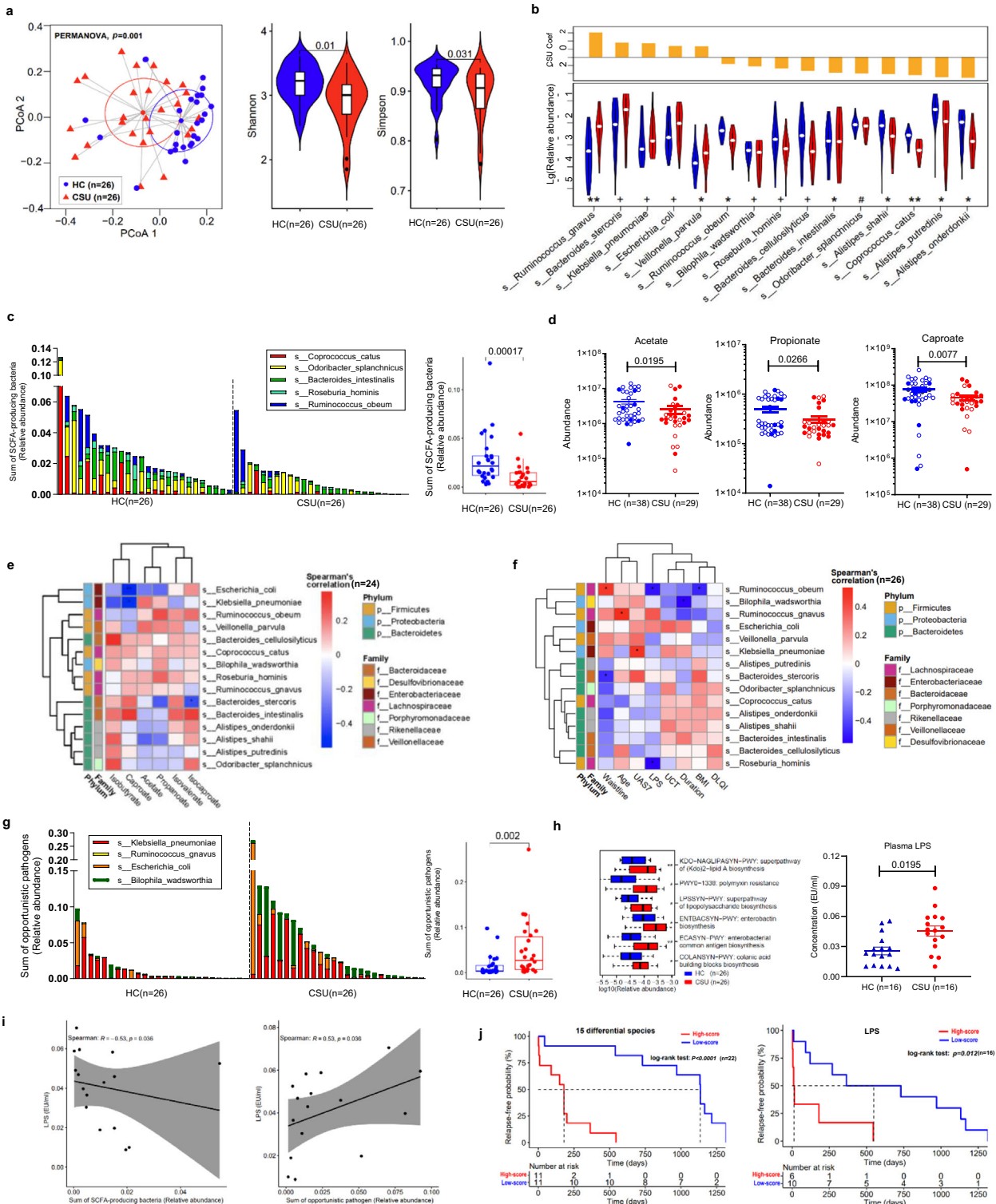

(*P* = 0.0123, Fig. 1j right panel), as well as 15 differential species (Fig. 1j left panel).

### The microbiome of CSU increases mast cell-driven skin inflammation, intestinal permeability, and blood LPS level

Next, we assessed the effects of the CSU gut microbiome on MC-driven skin inflammatory responses. To this end, we adoptively transferred the microbiome of CSU patients or HCs, by fecal microbial transplantation (FMT), to mice that had been subjected to antibiosis-induced bacterial depletion (Fig. 2a, Supplementary Fig. 3). As assessed

by metagenomic sequencing, CSU FMT but not HC FMT markedly changed the microbiome of recipient mice (Supplementary Fig. 4a, b), which acquired key CSU microbiome features including high levels of *Klebsiella pneumoniae* (Supplementary Fig. 4c). More importantly, CSU FMT, but not HC FMT, significantly increased passive cutaneous anaphylaxis (PCA) responses in recipient mice, with markedly increased vascular permeability (+70%, *P* < 0.0001, Fig. 2b) and MC degranulation (+240%, *P* < 0.0001, Fig. 2c), as well as significantly increased rates of MCs (CD45+CD117+FcεRIα+ cells)[39] in skin tissue detected by FACS (*P* < 0.001, Fig. 2d). In addition, CSU FMT, as

**Fig. 1 | CSU comes with reduced gut microbiome diversity, lower levels of SCFA-producing gut bacteria, higher levels of conditional pathogenic gut bacteria, decreased blood levels of SCFAs, and increased blood levels of LPS, which drive disease relapse. a** PCoA presents β-diversities based on Bray−Curtis dissimilarity determined by PERMANOVA, and α-diversity by Shannon and Simpson indices. HC vs CSU. **b** The 15 differential species identified by MaAsLin2 with BH correction. #$P$ < 0.2, +$P$ < 0.1, *$P$ < 0.05, **$P$ < 0.01. **c** Comparison of SCFA-producing bacteria between HC and CSU. **d** Abundance of acetate, propionate, and caproate through targeted metabolomics. **e** Spearman's correlation between SCFAs and differential species. **f** Spearman's correlation between differential species and clinical characteristics of CSU. The blocks in red or blue denote positive and negative correlation, respectively. The color gradient represents the strength of correlation. **g** Abundance of opportunistic pathogens was compared between HC and CSU. **h** LPS-related metabolic pathways and comparison of LPS level between HC and CSU. **i** Spearman's correlations between plasma LPS and sum of SCFA-producing bacteria or opportunistic pathogens. The line represents center of overall linear fit, with gray areas standing for the standard error of 95% confidence interval. **j** Relapse-free analysis based on the 15 differential species or plasma LPS by log-rank test. The data are presented as mean ± SEM. $P$ values were provided in Source Data File for **b**, **e**, and **f**. Two-tailed Wilcoxon test were used for **a** right, **c**, **g** right and **h** left. Two-tailed $t$ tests were used for **d**, **h** right. The shape and whiskers of violin plot respectively represent density distribution and overall range of the data in **a** and **b**. The center and bounds of box respectively represent median, the lower quartile (Q1) and the upper quartile (Q3) were in **a**–**c**, **g**, and **h**. The whiskers denote the lower quartile minus 1.5 times the IQR and the upper quartile plus 1.5 times the IQR in **c**, **g** and **h**. SCFAs short-chain fatty acids, HC healthy controls, CSU chronic spontaneous urticaria, BMI body mass index, UAS7 weekly urticaria activity score, DLQI Dermatology Life Quality Index, UCT Urticaria Control Test. Source data are provided as a Source Data file.

compared to HC FMT, increased intestinal permeability in recipient mice (+20%, $P$ < 0.01, Fig. 2e). To better characterize the effect of CSU FMT on mucosal integrity, we assessed the expression of epithelial tight junction molecules in the colon of recipient mice. CSU FMT down-regulated mRNA expression and protein levels of barrier function markers including ZO-1, MUC-2, occludin, TJP2, and CGN (Fig. 2f, g). Furthermore, CSU FMT up-regulated skin mRNA expression of *Il4*, *Il13*, *Tnfa*, and *Il10* mRNA ($P$ < 0.01, Fig. 2h), skin or plasma levels of IL4, IL13, and histamine (Fig. 2i, j), as well as plasma LPS levels (Fig. 2k).

### *Roseburia hominis* and short-chain fatty acid attenuate, whereas *Klebsiella pneumoniae* and LPS exacerbate mast cell-driven skin inflammation

Transplantation of mice with the SCFA producer *Roseburia hominis* (Fig. 3a), as verified by RT-PCR (Fig. 3b), markedly increased gut SCFA levels (butyrate, isobutyrate, isocaproate, and caproate), as assessed by targeted metabolomics (Fig. 3c), and ameliorated PCA responses including vascular permeability (Fig. 3d), MCs degranulation (Fig. 3e), population of MCs (CD45+CD117+FcεRIα+ cells) in skin tissue (Fig. 3f), and skin/plasma IL4, IL13, and histamine levels (Fig. 3g, h, i). Moreover, caproate inhibited IgE-mediated activation and cell mobility of MCs in vitro independent of GPR41 (G protein-coupled receptor 4; Fig. 4a, b; Supplementary Fig. 5), and its oral administration significantly suppressed IgE-mediated MCs activation in vivo (Fig. 4c–f).

In contrast, the adoptive transfer of *Klebsiella pneumoniae* to the gut of mice increased PCA responses and increased blood levels of LPS in recipient mice (Fig. 5a–e). LPS promoted IgE-induced activation in vitro (Fig. 5f) and increased PCA inflammatory responses in vivo (Fig. 5g–j). To further assess the mechanisms of LPS-facilitated MC activation, we used *Tlr4*−/− mice, which are deficient for the critical LPS receptor TLR4 (Fig. 6a; Supplementary Fig. 6a, b).

In contrast to WT mice, LPS treatment, in *Tlr4*−/− mice, did not increase PCA-induced vascular permeability (Fig. 6b), MC activation (Fig. 6c, d), and upregulation of IL4, IL13, and histamine in skin or plasma (Fig. 6e–g).

Also, MCs derived from *Tlr4*−/− mice (Supplementary Fig. 6b, c) showed lower levels of activation and cell mobility after incubation with LPS and IgE-mediated activation, as compared to MCs from *Tlr4*+/+ mice (Fig. 6h). Finally, we treated KPN-transplanted mice with cefoperazone/sulbactam (CPZ/SBT), which eradicates *Klebsiella pneumoniae*[40], markedly reduced PCA responses, plasma levels of LPS and altered the relative abundance of several differential bacteria (Fig. 7a–e, Supplementary Fig. 6d).

## Discussion

In this study, we show that the gut microbiome of patients with CSU is markedly different from that of healthy controls and that key differences such as reduced levels of SCFA-producers like *Roseburia hominis* and the increase of conditional pathogens like *Klebsiella pneumoniae* are linked to significantly increased IgE-mediated skin mast cell-driven inflammatory responses in vivo (Fig. 7f).

The imbalance of gut microbiota is held to contribute to the pathogenesis of several chronic diseases including inflammatory bowel diseases and metabolic disorders[41,42], however, the role of microbiota in CSU is unclear. In line with some but not all studies[35], our results demonstrate that the α-diversity of gut microbiota was reduced in CSU patients. More importantly, our study shows that CSU patients have a reduced abundance of beneficial SCFA-producing bacteria. Our findings are consistent with earlier reports of low levels of *Clostridium leptum*, *Ruminococcus bromii*, and *Bacteroides* in patients with CU[30,31] or CSU[35,43]. In addition, previous studies showed that the administration of SCFA-producing bacteria such as *Lactobacillus* or *Lactobacillus* combined with *Bifidobacterium* can reduce the risk of urticaria[26] and decrease CU disease severity[44,45], suggesting that SCFAs exert a critical role in protecting from CU including CSU.

SCFAs, key and beneficial metabolites of gut microbiota, have important physiological functions and maintain immunological homeostasis[46], by interacting with G protein coupled receptors and histone deacetylases[47]. In previous studies, high levels of dietary fiber increased blood propionic acid levels, which blocked allergic inflammation via GPCR41[48], and butyrate and propionate inhibited the activation of IgE or non-IgE-mediated mast cells activation[19,49]. In addition, butyrate and propionate may facilitate the differentiation of regulatory T cells[50,51] that control CD4(+) Th2 cells and their cytokines including IL4 and IL13, which contribute to the pathogenesis of CSU[52,53]. Our results suggest that SCFAs secure intestinal barrier function, at least in part, by promoting the expression of tight junction proteins such as claudin-1, ZO-1, and MUC2, thereby protecting from conditional pathogens[25]. These findings, together with the other results of our study, support the notion, that reduced levels of SCFA-producing gut bacteria and SCFA contribute to the pathogenesis of CSU. This is further supported by our findings on *Roseburia hominis*, a probiotic that generates SCFAs, such as acetate, propionate and butyrate, from dietary fiber and resistant starch[54]. *Roseburia hominis* has a critical role in maintaining intestinal epithelial barrier function and innate immune homeostasis[55], preventing the colonization with pathogenic bacteria, and enhancing nutrient absorption[56].

In our study, *Roseburia hominis* was negatively correlated with blood LPS levels of CSU patients. Furthermore, *Roseburia hominis* markedly increased gut SCFA levels and protected from skin mast cell degranulation in adoptively transferred mice. *Klebsiella pneumonia*, on the other hand, was positively correlated with CSU disease activity (UAS7), suggesting a contribution to the pathogenesis of CSU, possibly driven by the reduction of SCFA-producing bacteria and SCFA. It is well known that beneficial and harmful bacteria interact. SCFA-producing bacteria like *Lactobacillus reuteri* or *Clostridium orbiscindens*, suppress the proliferation of *Klebsiella pneumoniae*[57]. Thus, the reduction of

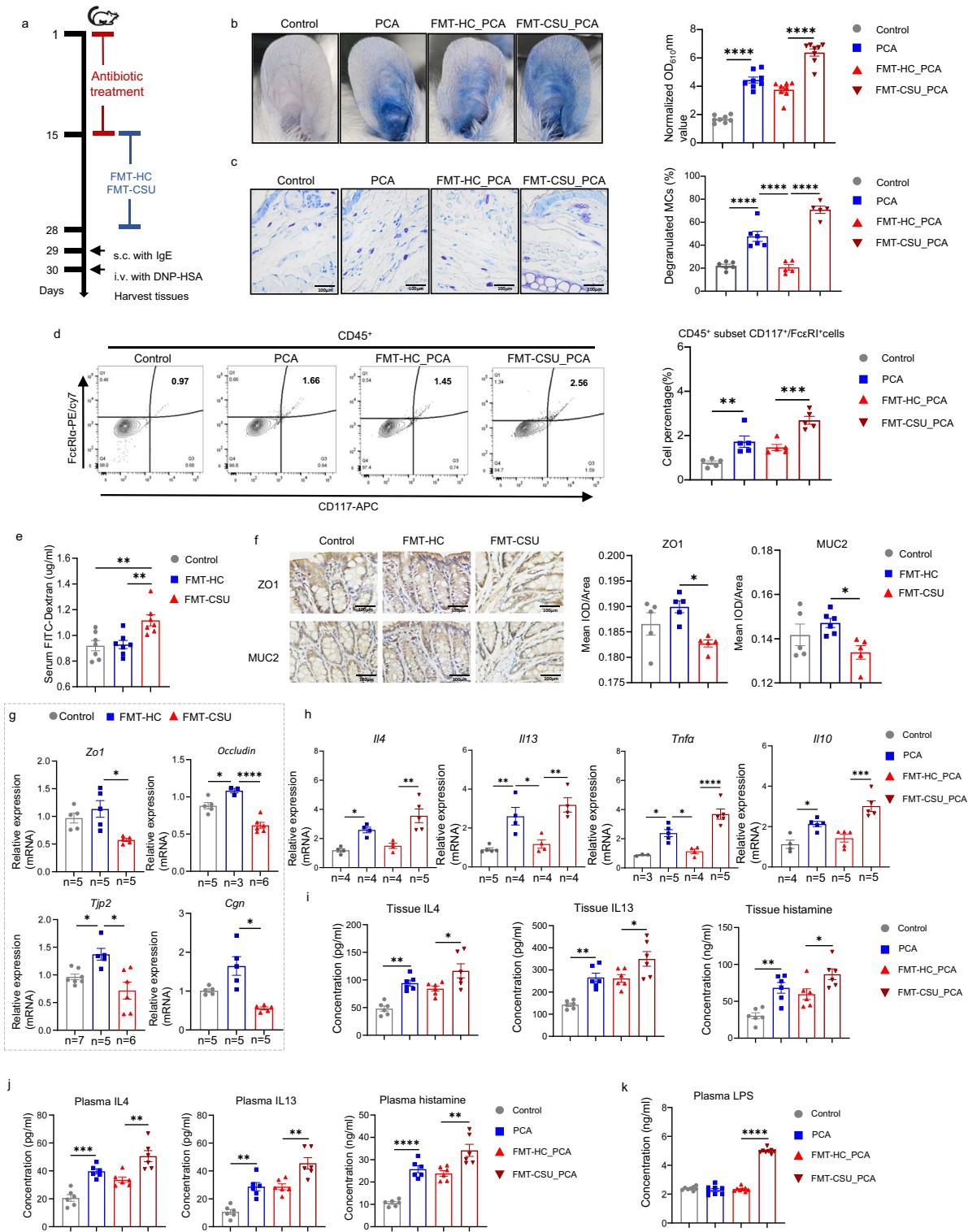

intestinal SCFA-producing bacteria in CSU is likely to be an important cause for enrichment of *Klebsiella pneumoniae*.

LPS, an important metabolite of gut microbiota, was significantly increased in CSU plasma and increased intestinal permeability in CSU FMT mice. LPS is synthesized by gram-negative bacilli including *Escherichia coli* and *Klebsiella pneumoniae*. Under normal circumstances, the intestinal epithelium is an effective barrier to LPS. In pathogenic states such as increased intestinal permeability or epithelial damage, LPS can shift from the intestinal cavity to the blood circulation[58,59]. Of note, intestinal permeability was shown to be

significantly increased in patients with CSU as assessed as measure with an in vivo triple glucose test[60]. LPS, via TLRs (TLR4), can activate MCs to produce various inflammatory mediators such as TNFa and IL13[61,62] and facilitates and enhances IgE-mediated MC degranulation[63], as confirmed by our results. It has to kept in mind, though, that the regulation of immunity by LPS is a complex process, with differences of *Bacteroides* LPS as compared to *Escherichia coli* LPS related to immunogenicity and endotoxin tolerance[64]. Furthermore, Sudhir et al. reported that low-dose LPS exposure enhanced allergic airway inflammation in OVA-sensitized mice, with higher levels of Th2

**Fig. 2 | The microbiome of CSU increases mast cell-driven skin inflammation, intestinal permeability, and blood LPS level. a** FMT mice model experimental design. **b** Representative ear images and the quantification of Evans blue dye ($n = 8$/group. Control vs PCA, $P < 0.0001$; FMT-HC_PCA vs FMT-CSU_PCA, $P < 0.0001$). **c** Representative images of MCs (×400 magnificent) and the percentages of degranulated MCs in mouse ear. Control ($n = 6$) vs PCA ($n = 6$), $P < 0.0001$; PCA vs FMT-HC_PCA ($n = 5$), $P < 0.0001$; FMT-HC_PCA vs FMT-CSU_PCA ($n = 5$), $P < 0.0001$. **d** Representative images of flow cytometry and the percentages of MCs in mouse skin ($n = 5$/group. Control vs PCA, $P = 0.0071$; FMT-HC_PCA vs FMT-CSU_PCA, $P = 0.0008$). **e** The amount of FITC-dextran detected in serum ($n = 7$/group. Control vs FMT-CSU, $P = 0.0053$; FMT-HC vs FMT-CSU, $P = 0.0075$). **f** Representative images of immunohistochemistry (×400 magnificent) and the quantification of ZO1 and MUC2 in mouse colon. FMT-HC vs FMT-CSU, $P = 0.0132$ (ZO1), and 0.0371 (MUC2). **g** mRNA expression of *Zo1*, *Occludin*, *Tjp2* and *Cgn* in mouse colon. FMT-HC vs FMT-CSU, $P = 0.0421$ (*Zo1*), <0.0001 (*Occludin*), $P = 0.0228$ (*Tjp2*), and 0.0215 (*Cgn*), respectively. Control vs FMT-HC, $P = 0.0263$ (*Occludin*), 0.0360 (*Tjp2*). **h** mRNA expression of *Il4*, *Il13*, *Tnfα* and *Il10* in ear skin (Control vs PCA, $P = 0.0388$, 0.0050, 0.0109, and 0.0191; FMT-HC_PCA vs FMT-CSU_PCA, $P = 0.0018, 0.0020, <0.0001$, and 0.0002 for *Il4*, *Il13*, *Tnfα* and *Il10*, respectively; PCA vs FMT-HC_PCA, $P = 0.0235$ (*Il13*), 0.0215 (*Tnfα*). **i** Levels of IL4, IL13, and histamine in mouse ear skin ($n = 6$/group. Control vs PCA, $P = 0.0019, 0.0033$, and 0.0032; FMT-HC_PCA vs FMT-CSU_PCA, $P = 0.0321, 0.0410$, and 0.0405 for IL4, IL13, and histamine, respectively) and **j** plasma detected by ELISA ($n = 6$/group. Control vs PCA, $P = 0.0003, 0.0012$, and <0.0001; FMT-HC_PCA vs FMT-CSU_PCA, $P = 0.0012, 0.0023$, and 0.0019 for IL4, IL13, and histamine, respectively). **k** Plasma LPS detected by ELISA ($n = 8$/group. FMT-HC_PCA vs FMT-CSU_PCA, $P < 0.0001$). *$P < 0.05$, **$P < 0.01$, ***$P < 0.001$, ****$P < 0.0001$, based on either one-way ANOVAs with Tukey's multiple comparisons test (**b**–**f**, **g**, *Zo1*/*Occludin*/*Tjp2*, **h**–**k**) or Brown–Forsythe and Welch ANOVA test (**g**, *CGN*). The data are presented as mean ± SEM of three independent experiments. Control: Solvents; PCA: Solvents+ IgE/DNP-HSA; FMT-HC_PCA: FMT from healthy control + IgE/DNP-HSA; FMT-CSU_PCA: FMT from CSU patients+ IgE/DNP-HSA. FMT fecal microbial transplantation, MCs mast cells. The numbers of biologically independent samples used in **g** and **h** is depicted in the figure. Source data are provided as a source data file.

cytokine production and reduced IFNγ production, whereas high-dose LPS contributed to the maintenance of allergic immune tolerance by elevating Tregs[65]. Further studies are needed to characterize, in detail, the mechanisms that lead to increased intestinal permeability in CSU and how the translocation of LPS to the bloodstream affects CSU diseases activity and duration. Interestingly, LPS can further damage the intestinal barrier and increase intestinal permeability, via effects on intestinal epithelial tight junction molecules including ZO-1 and occludin[66]. This may explain why our study showed that CSU patients with high LPS levels are more likely to have rapid relapse after cessation of controller therapy, suggesting that LPS may be a driver of CSU relapse. In our study, cefoperazone/sulbactam antibiotic treatment inhibited the promotion of IgE-mediated skin mast cell activation by the LPS producer *Klebsiella pneumoniae*, probably by its eradication, and this may be relevant for the beneficial effects of antibiotic treatment in some patients with CSU[67].

Our study has several strengths and some limitations. Its strengths include its comprehensive design, its complementary in vitro, ex vivo, and in vivo investigations, its use of metagenomics sequencing and metabolomics, correlating gut microbiome findings and clinical outcomes, and, most importantly, its proof-of concept strategy for demonstrating in vivo relevance of gut microbiome changes by adoptive transfer approaches including human-to-mouse fecal microbial transplantation.

As for limitations, our study did not assess the levels and effects of metabolic bacterial products other than LPS and SCFA, but there are many additional ones that may also play a role in CSU, including ligands of the Aryl hydrocarbon receptor (AhR), an important modulator of immune and inflammatory diseases[68]. Gut bacteria may also affect CSU by modifying primary bile acids and generating secondary bile acids (SBAs) such as deoxycholic acid, lithocholic acid and ursodeoxycholic acid, which are implicated in both innate and adaptive immune responses[69]. In autoimmune diseases, such as type 1 diabetes, there is a notable increase in the presence of bacteria that produce SBAs[70], and Elena et al. reported a higher abundance of bile acids (specifically taurocholate) in children suffering from asthma compared to the control group. The elevated levels of bile acids may be associated with the phenotypic expression and pathogenesis of asthma[71]. These and other potential mechanisms of microbiome effects on CSU should be investgated in future studies. Fecal microbial transfer to mice is a well-established and widely used model to study the pathogenic role of human gut microbes, for example in Alzheimer's disease[72], major depressive disorder[73], and chronic kidney disease[74]. However, mice and humans differ greatly in gastrointestinal physiology and extrapolating the results of mouse microbial transfer studies to the human system must be done with caution. Further studies including fecal transplantation studies in humans are needed to confirm and better characterize the role of gut microbiota in chronic spontaneous urticaria. Human fecal composition is only partially representative of the composition of the intestinal microbiome, and most nutrients are absorbed through the small intestine in the physiological state. Also, our study was monocentric and our results should be confirmed in multicenter studies across diverse patient populations. Finally, human-to-human fecal transfer studies were not performed, but are needed to demonstrate the clinical benefit of microbiome normalization in CSU patients.

Taken together, this study demonstrates that CSU comes with important and clinically relevant gut microbiome changes including lower levels of SCFA-producing bacteria and higher levels of bacterial LPS-producers. These gut microbiome alterations, by reducing and increasing SCFA and LPS levels, respectively, can facilitate MC-dependent skin inflammation and may drive the development of the signs and symptoms of CSU patients.

## Methods

### Ethics statement and consent to participate

This study was authorized by the ethics committee of Xiangya Hospital (approval No. 201904112) and complied with the declaration of Helsinki. All participants provided signed informed consent before joining the study. Sex has been consistent with gender information determined by self-reporting, appearance, and identity card. Animal experiments were performed in accordance with Chinese laws and regulations on laboratory animals: (1) Measures for ethical review of experimental animals(2003); (2)Welfare requirements for laboratory animals(2016); (3)Ethical Guidelines for the use of laboratory animals(2019). Animal experiments were approved by the ethics committee of experimental animal welfare of Central South University (approval No. 2021sydw0124). 3R principles (reducing, reusing, recycling) were applied to reduce the pain of animals as much as possible. All animal experiments were conducted in the Department of experimental zoology of Central South University, a specific pathogen-free facility. The feeding environment of mice was: temperature 21–23 °C, relative humidity 50–70%, 12 h light/12 h dark alternating environment, and free access to standard feed and sterilized water, sterilized mouse maintenance feed, specific pathogen free(SPF) grade(Cat#: MD17122, Jiangsu Medison Biopharmaceutical Co., Ltd) were provided for mice feeding.

### Study participants and conduct

Patients with CSU were recruited from the department of dermatology of Xiangya Hospital, Central South University, between Oct 2018 to Oct 2019. The diagnosis of CSU was carried out according to the international guideline for urticaria[75] (Supplementary Methods, Supplementary Fig. 1 and Supplementary Table 1 for details of participants'

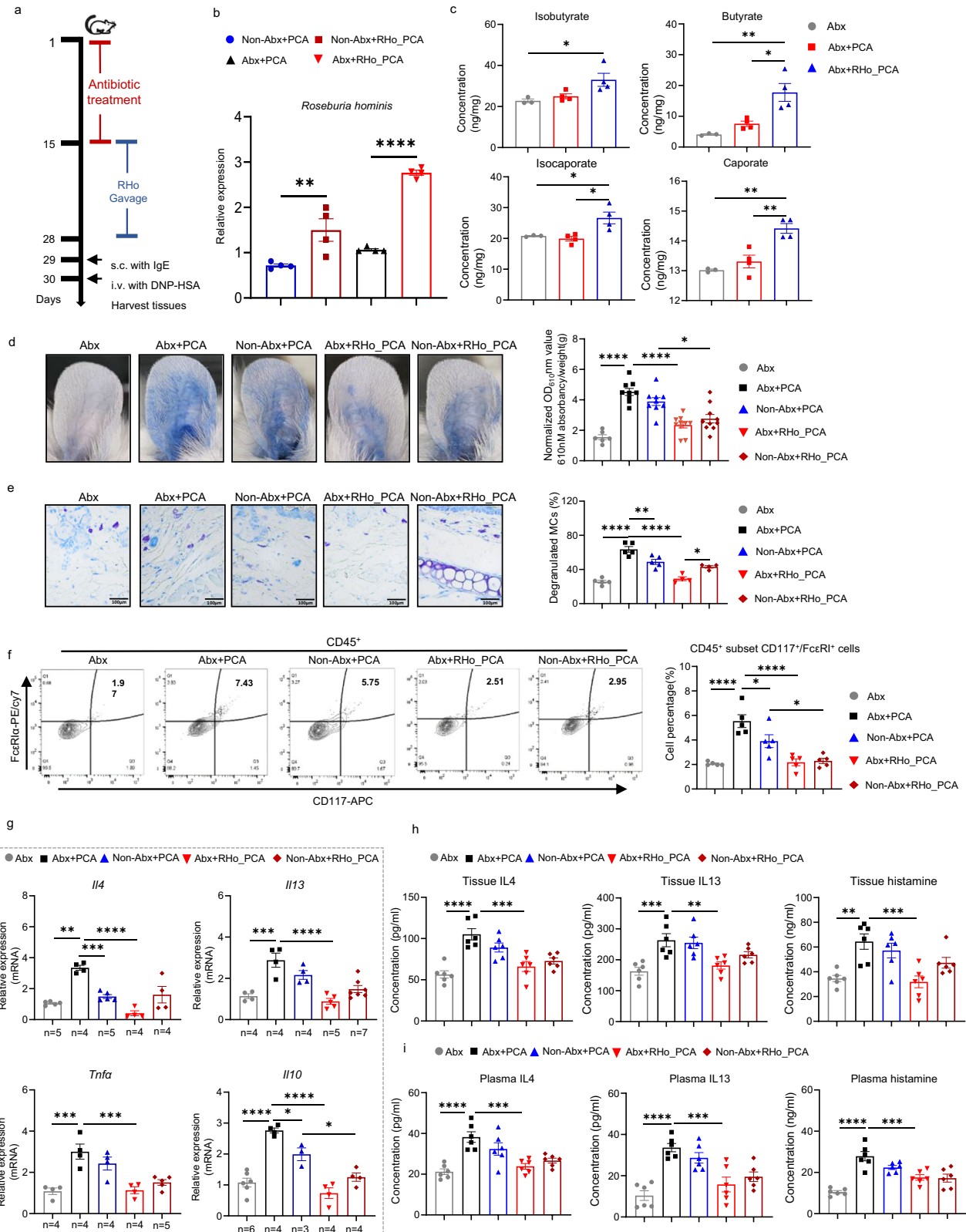

screening and research process). The inclusion criteria for patients with CSU were: 1) Aged 18–65; 2) No oral or topical antihistamines within one month before sample collection; 3) No antibiotics, prebiotics, probiotics, glucocorticoids, omalizumab and other drugs within three months before sample collection; 4) No consumption of yogurt, pickles and other fermented foods within three days before sample collection; 6) Living in Changsha city for more than 1 year

before sample collection. The exclusion criteria were: 1) Suffering from other subtypes of urticaria, such as symptomatic dermographism and acute urticaria, or allergic diseases; 2) With comorbid autoimmune disease (such as systemic lupus erythematosus, Sjogren's syndrome, thyroid problems, diabetes) and/or gastrointestinal symptoms; 3) Failure to collect fecal samples as required; 4) Lactation or pregnancy. The same criteria were applied for the recruitment of age- an

**Fig. 3 | Roseburia hominis attenuates mast cell-driven skin inflammation.**
**a** *Roseburia hominis* administration experimental design. **b** Abundance of *Roseburia hominis* by RT-PCR. ($n = 4$/group. Non-Abx+PCA vs Non-Abx+RHo_PCA, $P = 0.0050$; Abx+PCA vs Abx+RHo_PCA, $P < 0.0001$). **c** Abundance of SCFAs was tested by targeted metabolomics from cecal content of mice. Abx ($n = 3$) vs Abx+RHo_PCA ($n = 4$), $P = 0.0316$ (Isobutyrate), 0.0034 (butyrate), 0.0390 (isocaporate), and 0.0014 (caporate), respectively. Abx+PCA ($n = 4$) vs Abx+RHo_PCA ($n = 4$), $P = 0.0116$ (butyrate), 0.0140 (isocaporate), and 0.0036 (caporate), respectively. **d** Representative ear images and the quantification of Evans blue dye. Abx ($n = 6$) vs Abx+PCA ($n = 9$), $P < 0.0001$; Abx+PCA vs Abx+RHo_PCA ($n = 10$), $P < 0.0001$; Non-Abx_PCA ($n = 9$) vs Non-Abx+RHo_PCA ($n = 10$), $P = 0.0106$. **e** Representative images (×400 magnification) and the percentages of degranulated MCs in mouse skin. Abx ($n = 5$) vs Abx+PCA ($n = 5$), $P < 0.0001$; Abx+PCA vs Non-Abx+PCA ($n = 5$), $P = 0.0042$; Abx+PCA vs Abx+RHo_PCA ($n = 4$), $P < 0.0001$; Abx+RHo_PCA vs Non-Abx+RHo_PCA ($n = 4$), $P = 0.0157$. **f** Representative images of flow cytometry and the percentage of MCs in skin ($n = 5$/group. Abx vs Abx+PCA, $P < 0.0001$; Abx+PCA vs Non-Abx+PCA, $P = 0.0376$; Abx+PCA vs Abx+RHo_PCA, $P < 0.0001$; Non-Abx+PCA vs Non-Abx+RHo_PCA, $P = 0.0408$). **g** mRNA expression of *Il4*, *Il13*, *Tnfα* and *Il10* in mouse ear skin ($n$ for each group was provided in Source Data. Abx vs Abx+PCA: $P = 0.0011$ (*Il4*), 0.0002 (*Il13*), 0.0004 (*Tnfα*), and <0.0001 (*Il10*), respectively; Abx+PCA vs Non-Abx+PCA, $P = 0.0005$ (*Il4*), and 0.0333 (*Il10*), respectively. Abx+PCA vs Abx+RHo_PCA, $P < 0.0001$ (*Il4* and *Il13*), 0.0005 (*Tnfα*), and <0.0001 (*Il10*), respectively. Non-Abx+PCA vs Non-Abx+RHo_PCA, $P = 0.0434$ (*Il10*). **h** Levels of IL4, IL13, and histamine in mouse ear skin ($n = 6$/group. Abx vs Abx +PCA, $P < 0.0001$, 0.0009, and 0.0021; Abx+PCA vs Abx+RHo_PCA, $P = 0.0002$, 0.0077, and 0.0008, for IL4, IL13, and histamine, respectively) and **i** plasma detected by ELISA ($n = 6$/group. Abx vs Abx+PCA, all $P < 0.0001$; Abx+PCA vs Abx+RHo_PCA, $P = 0.0004$, 0.0005, and 0.0007, for IL4, IL13, and histamine, respectively). Abx: drinking water with Abx; Abx+PCA: drinking water with Abx+ IgE/DNP-HSA; Non-Abx+PCA: drinking water with solvents + IgE/DNP-HSA; Abx+RHo_PCA: drinking water with Abx + gavage with RHo + IgE/DNP-HSA; Non-Abx+RHo_PCA: drinking water with solvents+ gavage with RHo + IgE/DNP-HSA. One-way ANOVAs with Tukey's multiple comparisons test was used for **b**, **c**, **d**–**f** right, **g**–**i**. The data are presented as mean ± SEM of three independent experiments. *$P < 0.05$, **$P < 0.01$, ***$P < 0.001$, ****$P < 0.0001$. Abx antibiotics, RHo *roseburia hominis*. The number of biologically independent samples used in **g** is depicted in the figure. Source data are provided as a Source Data file (347/350).

sex-matched healthy controls (HCs) from the physical examination center of Xiangya Hospital of Central South University. Patients were assessed, for five years, for CSU recurrence, defined as the reappearance of CSU symptoms after complete remission and cessation of controller therapy[5].

## Collection and storage of human fecal and plasma samples

Fecal samples were collected within three minutes after defecation using sterile stool preservation tubes that contain stool preservation fluid and have a sterile spoon attached to the lid (Tinygene Biologicals, Shanghai, China, Product No. GWF01-A). Stool samples were stored at −80 °C for subsequent metagenomic sequencing and fecal microbiota transplantation (FMT). Blood was collected from the cubital vein with an anticoagulant tube containing ethylenediamine tetraacetic acid (Becton, Dickinson and company), rested at room temperature for 30 min, and centrifuged at 402 g for 10 min. Plasma was separated and stored at −80 °C.

## Metagenomic sequencing of human fecal samples and data analysis

Total genomic DNA was extracted from human fecal samples using MasterPure DNA Extraction Kit (Epicentre Co., Ltd., UK) following the manufacturer's instructions. The purity and concentration of total DNA were assessed by NanoDrop2000 and TBS-380, respectively. The quality of DNA was examined on 1% agarose gel. Genomic DNA was cut into fragments with an average of ~400 bp using a Bioruptor non-contact ultrasonic crusher (Diagenode Co., Ltd., Belgium) and paired-end sequencing was done using the Hiseq 2500 platform with Hiseq PE Cluster Kit V4 and Hiseq SBS Kit V4 250 Cycle Kit (Illumina Co., Ltd., USA).

The off-line raw data was quality controlled and cleaned by KneadData software (version 0.10.0), in which Trimmomatic (version 0.33) and Bowtie2 (version 2.2) were integrated to remove low-quality sequences and filter out human genome. Then, the clean data were subjected to taxonomic profiling and functional annotation by using HUMAnN2 software[76].

On the R platform (version 4.0.3), based on the relative abundance matrix at species level, α-diversities were evaluated by R package Vegan (version 2.5-7), such as Shannon and Simpson indexes, comparing the species richness between the two groups. Similarly, R package ade4 (version 1.7-18) and Vegan (version 2.5-7) were applied to evaluate β-diversity, comparing species communities between the two groups. To estimate the significance, permutational multivariate analysis of variance (PERMANOVA) was performed based on the Bray–Curtis distances. PCoA was performed based on

the Bray–Curtis dissimilarity matrix to visualize β-diversity. The β-diversity between the two groups at the functional gene families was assessed as well. Here, microbes detected in less than three samples were treated as noise, and microbes detected in at least half of the samples were defined as core microbes at multiple levels, including species, genus, and family. Then, core microbes were used for differential analysis by Wilcoxon test, and the cladogram of the core microbes with differential abundance was visualized by R package microbiomeViz (version 0.1.0). In addition, based on the relative abundance of the identified core species in HC and CSU patients, R package NetCoMi (version 1.0.2)[77] was used to construct the inter-relationship networks between core species, respectively, and the differential network between the two groups was constructed as well. Spearman's correlation between the differential species and clinical characteristics were analyzed by R package psych (version 2.1.9). Core species with differential abundance were used to construct classification models by R package random Forest (version 4.6-14) and for receiver operating characteristic (ROC) curves by R package pROC (version 4.6-14).

Similarly, after performing HUMAnN2 pipeline, functional annotation files for each sample were separately merged, including gene-families and pathabundance. Then, the two merged files were stratified to calculate relative abundance. Further, genes and pathways contributed by core microbes were extracted for subsequent analysis.

## Targeted metabolomics of short-chain fatty acids in human plasma samples

Stock solutions of the seven SCFAs standards were freshly and individually prepared in 50% aqueous acetonitrile. Then, a mixed standard solution containing 1 mM of each SCFA was made with the same solvent and diluted for calibration curves. For plasma samples, SCFAs from 20 ul plasma were extracted using precooled methanol/acetonitrile (2:1). After centrifugation, supernatant was used for derivatization with all calibration samples, and isotope-labeled internal standards of the seven SCFAs were used for normalization[78].

A Waters UPLC system (Waters, USA) was coupled to a QTRAP6500 (SCIEX, Canada) equipped with an ESI source and operated in the negative-ion mode. Chromatographic separations were performed on a Waters BEH C18 (2.1 × 50 mm, 1.7 mm) UPLC column, in which solvent A (water: formic acid; 100:0.01, v/v) and solvent B (acetonitrile: formic acid; 100:0.01, v/v) were used as the mobile phase for gradient elution. The column flow rate was 0.35 mL/min. All quantification data was processed using the MultiQuant 2.0 software (SCIEX, Canada).

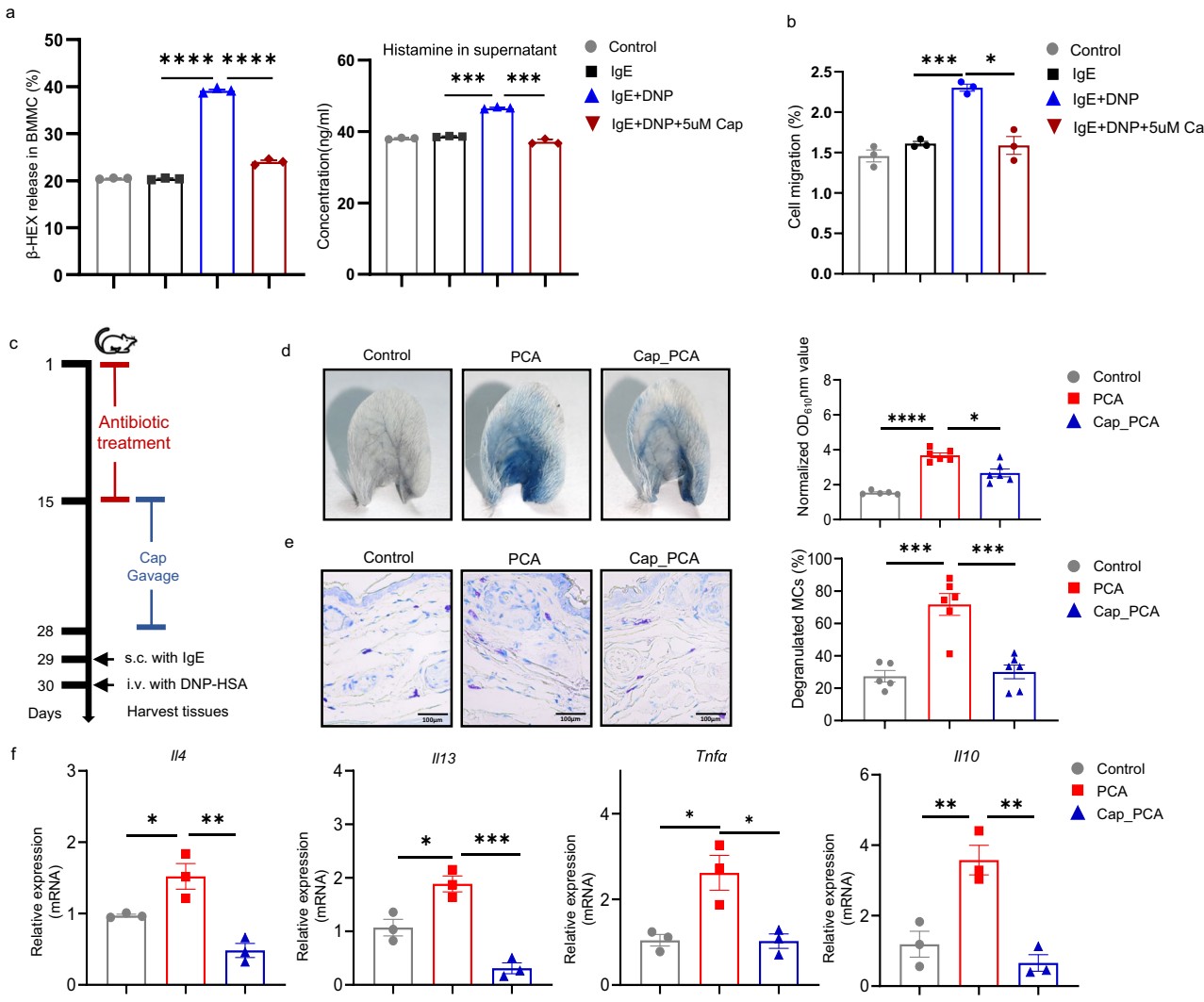

**Fig. 4 | SCFAs attenuate mast cell-driven skin inflammation and mast cell function. a** Release of β-hexosaminidase (left: $n = 3$/group. IgE vs IgE+DNP, IgE+DNP vs IgE+DNP+5uM Cap, both $P < 0.0001$) as well as concentration of histamine (right: $n = 3$/group. IgE vs IgE+DNP, $P = 0.0006$; IgE+DNP vs IgE+DNP+5uM Cap, $P = 0.0005$) from BMMCs. **b** Cell migration of BMMCs ($n = 3$/group, IgE vs IgE+DNP, $P = 0.0007$; IgE+DNP vs IgE+DNP+5uM Cap, $P = 0.0245$). **c** Caproate administration PCA mice model experimental design. **d** Representative ear images after PCA and the quantification of Evans blue dye. Control ($n = 5$) vs PCA ($n = 6$), $P < 0.0001$; PCA vs Cap_PCA ($n = 6$), $P = 0.0115$. **e** Representative images and the percentages of degranulated MCs in mouse ear. Control ($n = 5$) vs PCA ($n = 6$),

$P = 0.0001$; PCA vs Cap_PCA ($n = 6$), $P = 0.0001$. **f** mRNA expression of *Il4*, *Il13*, *Tnfα* and *Il10* in mouse ear skin ($n = 3$/group. Control vs PCA, $P = 0.0392, 0.0133, 0.0136$, and $0.0070$; PCA vs Cap_PCA, $P = 0.0020, 0.0005, 0.0128$, and $0.0025$ for *Il4*, *Il13*, *Tnfα*, and *Il10*, respectively). Control: gavage with solvents; PCA: gavage with solvents+IgE/DNP-HSA; Cap_PCA: gavage with caproate+IgE/DNP-HSA. One-way ANOVAs with Tukey's multiple comparisons test was used for **a**–**c**, **e** and **f** or Brown–Forsythe and Welch ANOVA test for **d**. The data are presented as mean ± SEM of three independent experiments. *$P < 0.05$, **$P < 0.01$, ***$P < 0.001$, ****$P < 0.0001$. Source data are provided as a Source Data file.

## Detection of LPS in human plasma samples

The concentration of LPS in plasma was determined by Bacterial endotoxin test kit according to the manufacturer's instructions (Dynamiker Biotechnology Co., Ltd, China).

## Preparation of human fecal suspension for FMT

Human fecal suspensions were prepared as follows[79]: briefly, 400 mg of fecal matter was dissolved in 10 ml of cysteine solution (0.1%), passed through a 100 μM screen, and the same volume of glycerol solution (30%). Equal volumes of fecal solutions from all patients were mixed, and the final fecal suspension was stored in 1.5 ml EP tubes at 4 °C.

## Fecal microbial transplantation in mice

Before performing FMT, all mice received antibiotic (Abx) treatment with ampicillin (1 mg/ml), colistin (1 mg/ml), and streptomycin (5 mg/ml) via sterile drinking water for 2 weeks. Human FMT was

administered to mice 2 days after the termination of Abx treatment. Mice were randomly divided into three groups (i.e., control, HC, CSU), and separately given, by oral gavage (200 ul), solvent of fecal suspension or fecal suspension from CSU patients or HC. The FMT was performed once a day for 2 weeks.

## Collection of mouse fecal samples and metagenomic sequencing

Feces from mice were collected before and after antibiotic treatment and after FMT or specific strain gavage. Before and after antibiotic treatment feces were assessed for changes by 16S rDNA sequencing. Post-FMT feces were subjected to metagenomic sequencing.

Total genomic DNA was extracted from mouse fecal samples using E.Z.N.A. Soil DNA Kit (Omega Bio-tek, Norcross, GA, U.S.) following the manufacturer's instructions. The purity and concentration of total DNA were detected by NanoDrop2000 and TBS-380, respectively.

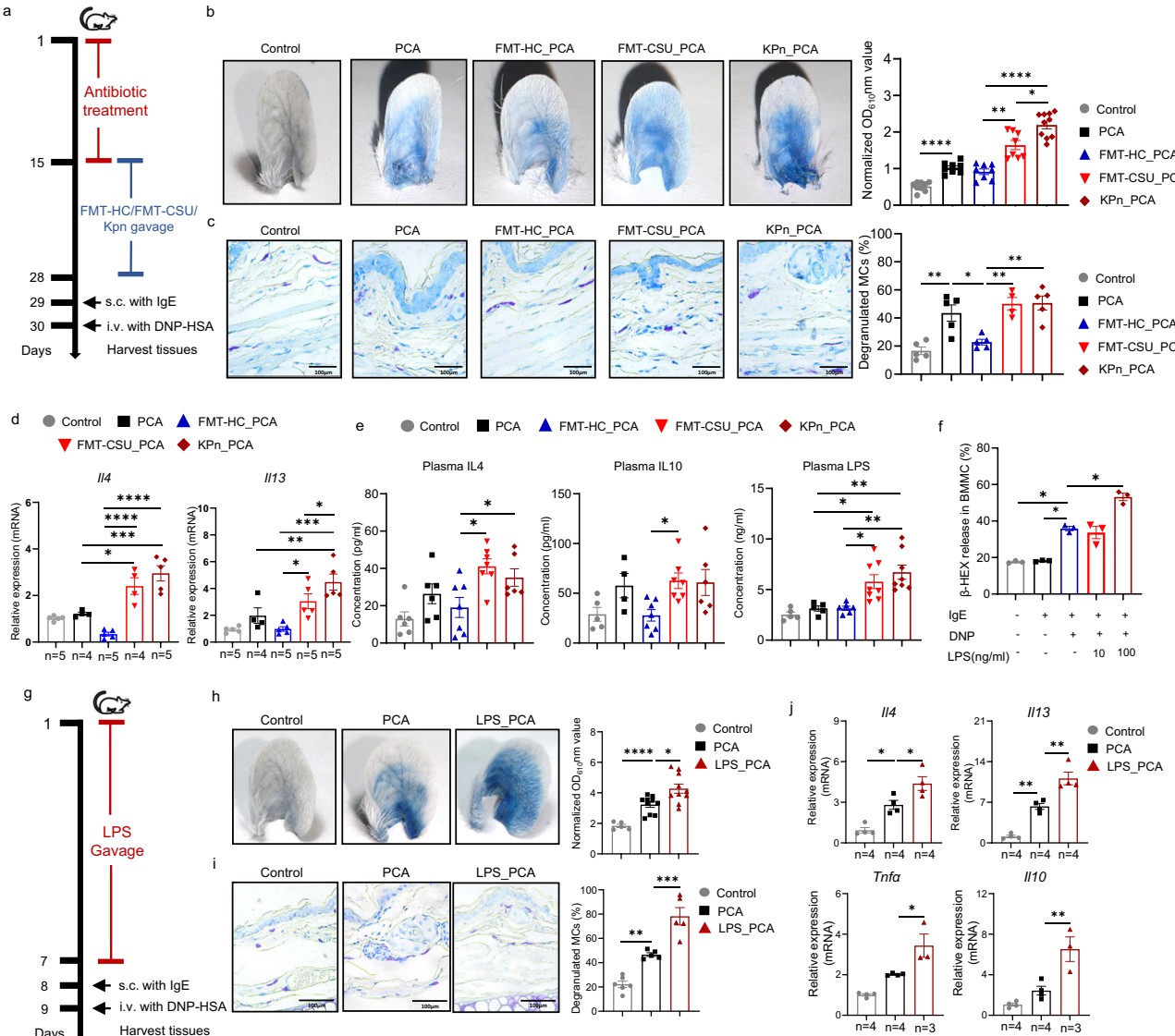

**Fig. 5 | Klebsiella pneumoniae and LPS exacerbate mast cell-driven skin inflammation. a** *Klebsiella pneumoniae* administration experimental design. **b** Representative ear images and quantification of Evans blue dye. Control (*n* = 9) vs PCA (*n* = 8), *P* < 0.0001; FMT-HC_PCA (*n* = 8) vs FMT-CSU_PCA (*n* = 8), *P* = 0.0027; FMT-HC_PCA vs KPn_PCA (*n* = 10), *P* < 0.0001; FMT-CSU_PCA vs KPn_PCA, *P* = 0.0314. **c** Representative images of toluidine blue staining (×400 magnification) and the percentages of degranulated MCs in mouse ear. Control (*n* = 5) vs PCA (*n* = 5), *P* = 0.0018; PCA vs FMT-HC_PCA (*n* = 5), *P* = 0.0174; FMT-HC_PCA vs FMT-CSU_PCA (*n* = 4); *P* = 0.0028, FMT-HC_PCA vs KPn_PCA (*n* = 5), *P* = 0.0012. **d** mRNA expression of *Il4, Il13* in ear skin. *Il4*: PCA vs FMT-CSU_PCA, *P* = 0.0150; PCA vs KPn_*PCA*, *P* = 0.0002; FMT-HC_PCA vs FMT-CSU_PCA, *P* < 0.0001. FMT-HC_PCA vs KPn_PCA, *P* < 0.0001. *Il13*: PCA vs KPn_PCA, *P* = 0.0088; FMT-HC_PCA vs KPn_PCA, *P* = 0.0002; FMT-HC_PCA vs FMT-CSU_PCA, *P* = 0.0247. **e** Levels of IL4, IL10, LPS in mouse plasma. IL4: Control (*n* = 6) vs FMT-CSU_PCA (*n* = 7), *P* = 0.0250; PCA (*n* = 7) vs FMT-CSU_PCA (*n* = 7), *P* = 0.0139. IL10: FMT-HC_PCA (*n* = 6) vs FMT-CSU_PCA (*n* = 8), *P* = 0.0371. LPS: PCA (*n* = 5) vs FMT-CSU_PCA (*n* = 8), *P* = 0.0426; PCA vs KPn_PCA (*n* = 8), *P* = 0.0080; FMT-HC_PCA (*n* = 6) vs FMT-CSU_PCA (*n* = 8), *P* = 0.0469; FMT-HC_PCA vs KPn_PCA (*n* = 8), *P* = 0.0099. **f** β-hexosaminidase release in BMMCs (*n* = 3/group. Control vs IgE+DNP, *P* = 0.0170; IgE vs IgE+DNP, *P* = 0.0170; IgE+DNP vs IgE+DNP + 100 ng/ml LPS, *P* = 0.0251). **g** LPS administration experimental design. **h** Representative ear images and the quantification of Evans blue dye. Control (*n* = 5) vs PCA (*n* = 10), *P* < 0.0001; PCA vs LPS_PCA (*n* = 10), *P* = 0.0260. **i** Representative images of toluidine blue staining (×400 magnification) and the percentages of degranulated MCs in ear. Control (*n* = 6) vs PCA (*n* = 5), *P* = 0.0036; PCA vs LPS_PCA (*n* = 5), *P* = 0.0006. **j** mRNA expression of *Il4, Il13, Tnfα* and *Il10* in mouse ear skin. Control vs PCA, *P* = 0.0145 (*Il4*), and 0.0015 (*Il13*). PCA vs LPS_PCA, *P* = 0.0360 (*Il4*), 0.0024 (*Il13*), 0.0147 (*Tnfα*), and 0.0050 (*Il10*). Control: Solvents; FMT-HC_PCA: FMT from healthy control + IgE/DNP-HSA; FMT-CSU_PCA: FMT from CSU patients + IgE/DNP-HSA; KPn_PCA: Gavage with KPn+IgE/DNP-HSA; LPS_PCA: Gavage with LPS+IgE/DNP-HSA. One-way ANOVAs with Tukey's multiple comparisons test One-way ANOVA was used for **a–d**, **e** IL4/LPS, **f–j**. Brown–Forsythe and Welch ANOVA test was used for **e** IL10. The data are presented as mean ± SEM of three independent experiments. **P* < 0.05, ***P* < 0.01, ****P* < 0.001, *****P* < 0.0001. KPn *Klebsiella pneumonia,* LPS lipopolysaccharides. The number of biologically independent samples used in **d** and **j** is depicted in the figure. Source data are provided as a Source Data file.

The quality of DNA was examined on 2% agarose gel. Genomic DNA was cut into fragments with an average of ~400 bp using Covaris M220 (Gene Company Limited, China). Then, the v3-v4 region of 16S rRNA gene was amplified with specific primers with connector markers, and paired-end sequencing was done using the Illumina NovaSeq6000 platform and NovaSeq Reagent Kits (Illumina Co., Ltd., USA).

Off-line reads are analyzed on the Majorbio cloud platform (cloud. Majorbio. Com), as described below. BWA (version 0.7.9a) was used to remove human DNA sequences, and fastp (version 0.20.0) was used to obtain high quality pair end reads by removing reads of <50 bp and adapter sequences. MEGHIT (version 1.1.2)[80] was used to assemble the optimized sequence and screen countings

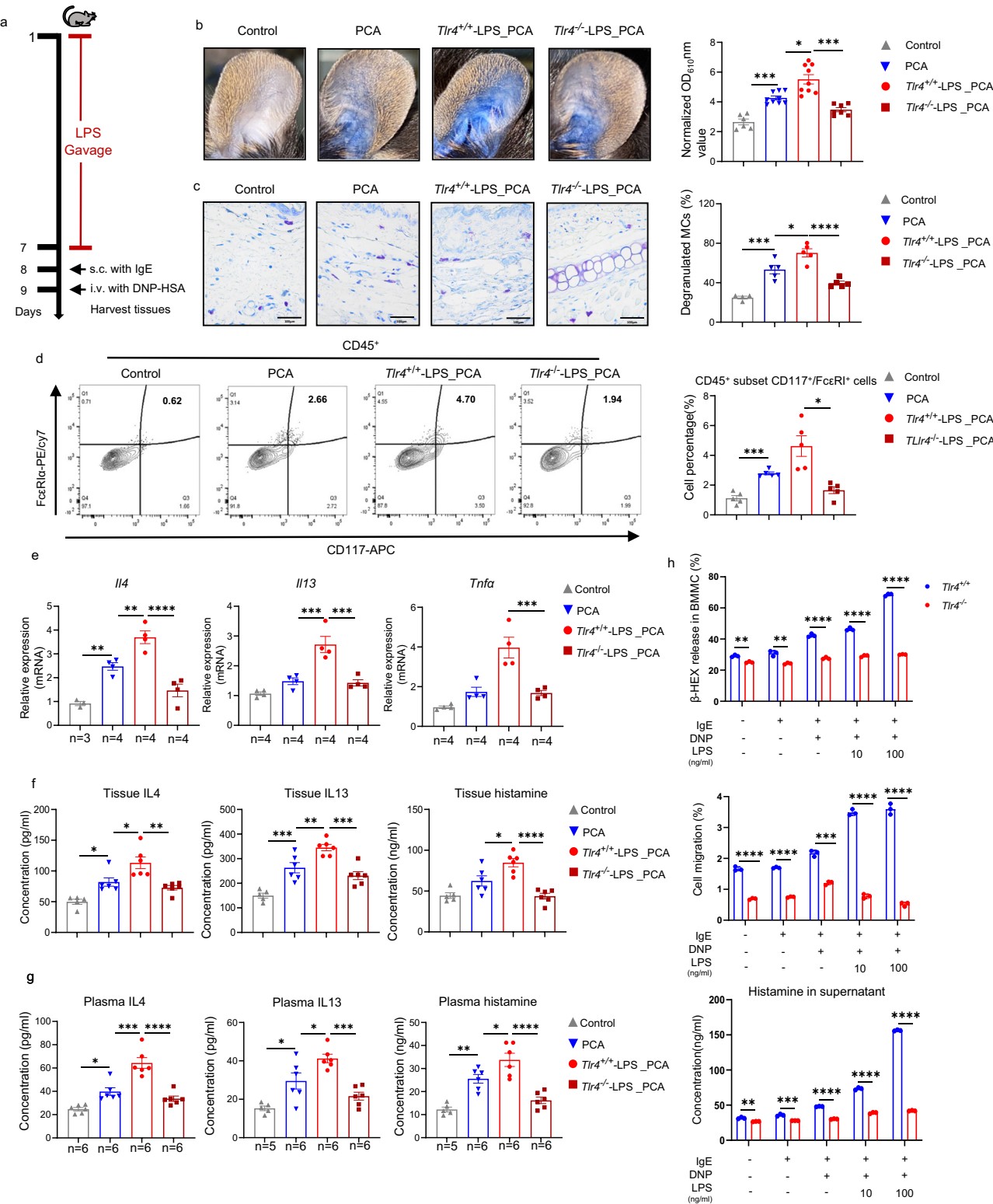

of ≥300 bp. Then, metagene (http://metagene.cb.k.u-tokyo.ac.jp/) was used to predict ORF and select genes of ≥100 bp. CD-HIT (version 4.6.1)[81] was used to cluster the gene sequences (parameters: 90% identity and 90% coverage), and take the longest sequences as the representative sequence to construct a non-redundant gene set. SOAPaligner (version 2.21) was used to compare the high-quality reads in each sample with the non-redundant gene set (95% identity) to obtain the gene abundance in the corresponding sample. Diamond (version 0.8.35) was used to compare the non-

redundant gene set with the NR database (the blastp alignment parameter is set to the expected value E-value of 1e−5) to obtain the taxonomic annotation of the species and calculate the abundance of the species.

Based on the relative abundance at species level, α-diversity (i.e., Shannon and Simpson indexes) and β-diversity were estimated with R package ade4 (version 1.7-18) and Vegan (version 2.5-7). Core species were selected and used for differential analyses between mice in different groups by Wilcoxon test.

**Fig. 6 | TLR4 plays an important role in LPS exacerbated mast cell-driven skin inflammation. a** LPS administration on *Tlr4*⁺/⁺ and *Tlr4*⁻/⁻ mice experimental design. **b** Representative ear images and the quantification of Evans blue dye. Control (*n* = 6) vs PCA (*n* = 9), *P* = 0.0004; PCA vs *Tlr4*⁺/⁺-LPS_PCA (*n* = 9), *P* = 0.0203; *Tlr4*⁺/⁺-LPS_PCA vs *Tlr4*⁻/⁻-LPS_PCA (*n* = 7), *P* = 0.0006. **c** Representative images of toluidine blue staining (×400 magnification) and the percentages of degranulated MCs in mouse ear skin. Control (*n* = 4) vs PCA (*n* = 5), *P* = 0.0002; PCA vs *Tlr4*⁺/⁺-LPS_PCA (*n* = 5), *P* = 0.0110; *Tlr4*⁺/⁺-LPS_PCA vs *Tlr4*⁻/⁻-LPS_PCA (*n* = 5), *P* < 0.0001. **d** Representative images of flow cytometry and the percentage of activated MCs in mouse ear skin (*n* = 5/group. Control vs PCA, *P* = 0.0009; *Tlr4*⁺/⁺-LPS_PCA vs *Tlr4*⁻/⁻-LPS_PCA, *P* = 0.0439). **e** mRNA expression of *Il4*, *Il13* and *Tnfα* in mouse ear skin (*Il4*: Control vs PCA, *P* = 0.0030; PCA vs *Tlr4*⁺/⁺-LPS_PCA, *P* = 0.0094; *Tlr4*⁺/⁺-LPS_PCA vs *Tlr4*⁻/⁻-LPS_PCA, *P* < 0.0001. *Il13*: PCA vs Tlr4⁺/⁺-LPS_PCA, *P* = 0.0007; *Tlr4*⁺/⁺-LPS_PCA vs *Tlr4*⁻/⁻-LPS_PCA, *P* = 0.0005. *Tnfα*: *Tlr4*⁺/⁺-LPS_PCA vs *Tlr4*⁻/⁻-LPS_PCA, *P* = 0.0007. **f** Levels of IL4, IL13, histamine in mouse ear skin (Control, *n* = 5; PCA, *Tlr4*⁺/⁺-LPS_PCA, *Tlr4*⁻/⁻-LPS_PCA, *n* = 6/group. Control vs PCA, *P* = 0.0181 (IL4), and 0.0006 (IL13); PCA vs Tlr4⁺/⁺-LPS_PCA, *P* = 0.0152 (IL-4), 0.0076 (IL-13), and 0.0166 (histamine); *Tlr4*⁺/⁺-LPS_PCA vs *Tlr4*⁻/⁻-LPS_PCA, *P* = 0.0017 (IL4), 0.0003 (IL13), and <0.0001 (histamine), respectively) and **g** plasma detected by ELISA (Control, *n* = 6 for IL4, *n* = 5 for IL13 and histamine; PCA, *Tlr4*⁺/⁺-LPS_PCA, *Tlr4*⁻/⁻-LPS_PCA, *n* = 6/group. Control vs PCA, *P* = 0.0132 (IL4), 0.0105 (IL13), 0.0011 (histamine), respectively; PCA vs Tlr4⁺/⁺-LPS_PCA, *P* = 0.0001(IL-4), 0.0297(IL-13), and 0.0378 (histamine), respectively; *Tlr4*⁺/⁺-LPS_PCA vs *Tlr4*⁻/⁻-LPS_PCA, *P* < 0.0001 (IL4), 0.0003 (IL13), and <0.0001 (histamine), respectively. **h** β-hexosaminidase release, cell migration and concentration of histamine in BMMCs from *Tlr4*⁺/⁺ and *Tlr4*⁻/⁻ mice treated with LPS (*n* = 3/group). *Tlr4*⁺/⁺IgE+DNP vs *Tlr4*⁻/⁻IgE+DNP, *Tlr4*⁺/⁺IgE+DNP + 10 ng/ml LPS vs *Tlr4*⁻/⁻IgE+DNP + 10 ng/ml LPS, *Tlr4*⁺/⁺IgE+DNP + 100 ng/ml LPS vs *Tlr4*⁻/⁻ IgE+DNP + 100 ng/ml LPS, all *P* < 0.0001 for β-hexosaminidase and histamine release; *Tlr4*⁺/⁺IgE+DNP vs *Tlr4*⁻/⁻IgE+DNP, *P* = 0.0001 for cell migration. One-way ANOVAs with Tukey's multiple comparisons test was used for **b**–**d** right, **e**–**g**. Two tailed *t* test was used for **h**. The data are presented as mean ± SEM of three independent experiments. **P* < 0.05, ***P* < 0.01, ****P* < 0.001, *****P* < 0. 0001.The number of biologically independent samples used in **e** and **g** is depicted in the figure. Source data are provided as a Source Data file.

## Analysis of 16S rDNA sequencing data of experimental mice fecal samples

The assembly and quality control of paired data were carried out to obtain effective sequences by removing the barcodes and primer sequences and overlapping sequences. Then, the effective sequences obtained above were used for OTU clustering and species annotation by Uparse software. Sequences with the similarity of ≥97% were considered as the same OTU, and the sequence with the highest frequency was taken as the representative sequence of OTU. Each representative sequence was used for species annotation by using Silva database based on Mothur algorithm. The OTU abundance was standardized by the number of sequences corresponding to the samples with the least sequences.

Based on the standardized OTU abundance, the α-diversities in each sample, e.g., Shannon index, was evaluated by qiime software (version 1.7.0) and displayed by R software package (version 2.15.3). Based on Bray–Curtis distance with qiime software (version 1.9.1), the β-diversity between CSU and HC groups was calculated and displayed on principal coordinate analysis (PCoA). In addition, the LEfSe software (version 1.0) was used for linear discriminant analysis to find biomarkers with statistical differences between groups.

## Targeted metabolomics of short-chain fatty acids in mouse feces

Targeted gas chromatography-mass spectrometry (GC-MS) analysis was performed to measure short-chain fatty acids (SCFAs). Mouse fecal sample (25 mg) were placed in 2 mL grinding tubes with 500 μL water containing 0.5% phosphoric acid. Samples were frozen and ground at 50 Hz for 3 min, repeated twice, followed by ultrasound for 10 min and centrifugation at 4 °C and 13000 g for 15 min. Supernatant (400 μL) was transferred to a 1.5 mL centrifuge tube, and N-butanol solvent (0.2 mL) containing internal standard 2-ethylbutyric acid (10 μg/mL) was added. After vortex for 10 s, ultrasound at low temperature for 10 min, and centrifugation at 4 °C and 13000 g for 5 min, the supernatant was transferred to sample vials for analysis.

The analysis was performed using an Agilent 8890B gas chromatography coupled to an Agilent 5977B/7000D mass selective detector with an inert electron impact (EI) ionization source and ionization voltage was 70 eV (Agilent, USA). Analyte compounds were separated with a HP-FFAP (30 m × 0.25 mm × 0.25 μm) capillary column, using 99.999% helium as a carrier gas at a constant flow rate (1 mL/ min). The GC column temperature was programmed to hold at 80 °C and rise to 120 °C at a rate of 40 °C per minute, then rise to 200 °C at a rate of 10 °C per minute, and then hold at 230 °C for 3 min. The injection volume of samples was 1 μL and introduced in splitting mode (10:1) with the inlet temperature of 180 °C. The ion sources temperature was 230 °C and the quadrupole temperature was 150 °C.

The scanning mode was Selected ion Monitor. Compounds were identified and quantified by software of Masshunter (v10.0.707.0, Agilent, USA). The mass spectrum peak area of the analyte was used as the ordinate and the concentration of the analyte as the abscissa to draw a linear regression standard curve for sample concentration calculation: the mass spectrum peak area of the sample analyte was substituted into the linear equation to calculate the concentration result.

The default parameters of Masshunter quantitative software (Agilent, USA, version number: v10.0.707.0) were used to automatically identify and integrate each ion fragment of the target short-chain fatty acid, and assist manual inspection.

## Assessment of intestinal permeability in mice

In vivo intestinal permeability was assessed with 4000 Da FITC-dextran (46944, Sigma-Aldrich)[66]. In brief, mice were fasted for more than 6 h and were given 4000 Da FITC dextran working solution (200 ul, 50 mg/ml) by gavage. After 2 h, whole blood was collected and centrifuged for serum. The serum was diluted with the same amount of PBS and the absorbance of it was measured by spectrophotometer. The standard curve was obtained with different diluted concentrations of working solution and corresponding absorbance values.

## Passive cutaneous anaphylaxis (PCA) in mice

PCA was induced as follows[82]: briefly, both ears of mice were intradermally injected with 10 ug/ml IgE (25ul, D8406, Sigma-Aldrich). The tail vein, 16 h later, was injected with 1 mg/ml DNP-HSA (100ul, D-5059, Biosearch Technologies). After 30 min, a full thickness biopsy of the right ear was fixed and embedded for toluidine blue staining. Evans blue was extracted from the remaining skin tissue of the right ear at 65 °C for 16 h, and its absorbance was measured at 610 nm. After 12 h, blood was collected, and the concentration of IL4, IL13, and IL10 in plasma/ear tissue was determined by ELISA according to the manufacturer's instructions (Jianglai Biotechnology Co., Ltd., China). The concentration of LPS in plasma and histamine in plasma/ear tissue was determined by ELISA (USCN Life Science and Technology, China). The left ear tissue was frozen in liquid nitrogen and stored at −80 °C for subsequent qPCR detection.

## Bacterial strains and culture conditions

*Klebsiella pneumoniae* (BNCC 102997 = ATCC 10031) was incubated using nutrient broth medium for 16–18 h at 37 °C in a constant temperature shaker at 220 rpm. *Roseburia hominis* (DSM 16839) was incubated using PYG MEDIUM (modified) at 37 °C for 16–18 h in an anaerobic incubator. Cell pellets were washed twice and adjusted to $1.0 \times 10^9$ colony-forming units (CFU)/mL.

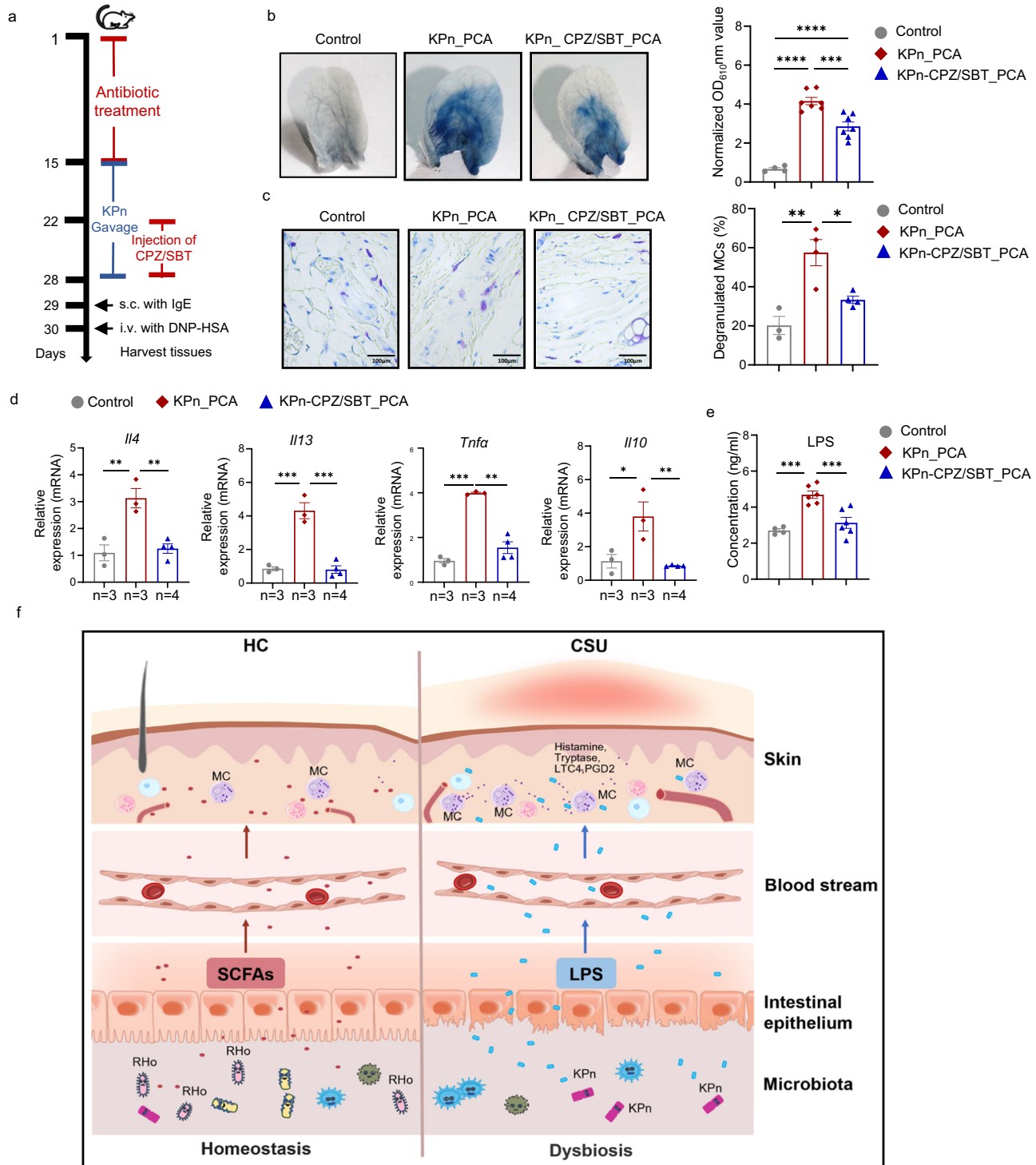

**Fig. 7 | Cefoperazone/sulbactam reduces mast cell activation in *Klebsiella pneumoniae*-exacerbated PCA reaction. a** Experimental design of cefoperazone/sulbactam treated *Klebsiella pneumoniae* transplanted mice. **b** Representative ear images and the quantification of Evans blue dye. Control (*n* = 4) vs KPn_PCA (*n* = 7), *P* < 0.0001; KPn_PCA vs KPn_CPZ/SBT_PCA (*n* = 7), *P* = 0.0006. **c** Representative images of toluidine blue (×400 magnification) and the percentages of degranulated MCs in ear. Control (*n* = 3) vs KPn_PCA (*n* = 4), *P* = 0.0021; KPn_PCA vs KPn_CPZ/SBT_PCA (*n* = 4), *P* = 0.0164. **d** mRNA expression of *Il4*, *Il13*, *Tnfα* and *Il10* in mouse ear skin. Control vs KPn_PCA, *P* = 0.0034 (*Il4*), 0.0003 (*Il13*), 0.0003 (*Tnfα*), and

0.0185 (*Il10*), respectively; KPn_PCA vs KPn_CPZ/SBT_PCA, *P* = 0.0037 (*Il4*), 0.0002 (*Il13*), 0.0064 (*Tnfα*), and 0.0080 (*Il10*), respectively. **e** Plasma level of LPS detected by ELISA. Control (*n* = 4) vs KPn_PCA (*n* = 6), *P* = 0.0003; KPn_PCA vs KPn_CPZ/SBT_PCA (*n* = 6), *P* = 0.0010. **f** Schematic of gut microbiota facilitating chronic spontaneous urticaria. One-way ANOVAs with Tukey's multiple comparisons test was used for **b** right, **c** right, **d** and **e**. The data are presented as mean ± SEM of three independent experiments. **P* < 0.05, ***P* < 0.01, ****P* < 0.001, *****P* < 0.0001. CPZ/SBT cefoperazone/sulbactam. The number of biologically independent samples used in **d** is depicted in the figure. Source data are provided as a Source Data file.

## Animals and experimental design

Female BALB/c mice with specific pathogen free (SPF), aged 6–8 weeks and weighing about 20–23 g, came from Hunan SJA Laboratory Animal Co. Ltd. There were no more than 5 mice in each cage. After 1 week of habituation to the laboratory environment, mice were subjected to animal experiments. Jie Li and Cong Peng were aware of the group allocation at the different stages of the experiment.

Mice were randomly divided into five groups and pre-treated with a cocktail of broad-spectrum antibiotics for 2 weeks, including ampicillin (1 mg/ml), colistin (1 mg/ml), and streptomycin (5 mg/ml). The KPn group then received *Klebsiella pneumoniae* (200 μL/mouse at $1.0 \times 10^9$ CFU/mL) by oral gavage daily for 2 weeks. In parallel, the control group received 200 μL PBS by oral gavage. On day 29, fecal samples of mice were collected into a sterile tube, snap-frozen in liquid nitrogen, and stored at −80 °C until further analysis, and mice were subjected to PCA.

Given Toll-like receptor 4 (TLR4) plays an important role in LPS-induced biological function, therefore, we investigated the effcts of TLR4 on LPS facilitating MCs activation. *Tlr4*[-/-] C57BL/6 mice and *Tlr4*[+/+] C57BL/6 mice[83] (kindly gifted by Dr. Ben Lu, Laboratory of Sepsis Translational Medicine, Hunan, China) in the LPS_PCA group gavaged with LPS (200 ul, Sigma, St. Louis, USA) at a dose of 1 mg/kg daily for one week (control mice received 200 ul PBS).

Animals were pre-treated with a cocktail of broad-spectrum antibiotics for 2 weeks and then received *Klebsiella pneumoniae* for 2 weeks (Same as above). On day 22–28, mice were injected intraperitoneally with 100 uL cefoperazone/sulbactam (1:1) at concentration of 300 mg/kg/d twice daily for last one week. Control groups were received with 100 ul sterile saline.

To evaluate *Roseburia hominis* effects on PCA reaction, live *Roseburia hominis* ($2 \times 10^8$ CFU, 200 ul) were gavaged to antibiotic pre-treated mice daily for 2 weeks. Control groups were received 200 ul PBS.

For oral caporate supplementation, mice in the Caproate_PCA group received sodium caproate (400 mg/kg, 200 ul) by oral gavage daily from week 2 to week 4 (control group received 200 ul sterile saline).

## Total RNA extraction, reverse transcription and qPCR in mice

Total RNA was extracted by using total RNA Isolation Kit (Trizol, Bio Teke, China) and reverse transcribed by using HifairTM III 1st Strand cDNA Synthesis SuperMix for qPCR (gDNA digester plus, Yeasen, China) according to the reagent manufacturer's instructions. qPCR was performed at 50 °C for 2 min and 95 °C for 10 min, followed by 40 PCR cycles (95 °C for 15 s and 60 °C for 1 min), and then at 95 °C for 15 s and 60 °C for 1 min to obtain the dissolution curve and terminate the reaction. The primers used in this experiment are shown in Supplementary Table 3 (Sangon Biotech, China).

## Histology and immunohistochemistry (IHC)

Immunohistochemical analysis for ZO1 and MUC2 was performed on 4-μm-thick paraffin-embedded sections from colon of control and FMT mice. Briefly, sections were blocked with normal sheep serum albumin (ZSGB-BIO, China) and the antibodies, ZO1 (ABclonal Technology, China, dilution 1:100, Cat#, A0659, Lot, 5500020603) and MUC2 (ABclonal Technology, China, dilution 1:50, Cat#, A4767, Clone, ARC1012) were added and incubated for 12 h at 4 °C. Antibody dilution buffer was used as a negative control. Using the PV-9000 universal two-step detection kit (ZSGB-BIO, China) and DAB chromogenic agent (ZSGB-BIO, China), the sections were stained with hematoxylin (ZSGB-BIO, China). Photomicrographs were obtained from three random high-power microscopic fields (×400 magnification) and assessed by Image-Pro-Plus software (version 6.0). Mean density was determined based on the rate of integral optical density sum and area.

## Flow cytometry (FCM)

Single-cell suspensions were prepared from ear tissues, 1 μl of prepared Zombie Aqua Fixable Viability Kit (anti-BV510, BioLegend, USA) was added and incubated for 15 min at room temperature, and 1 μl of prepared surface antibody [Anti-mouse CD45-APC/cy7, (BioLegend; Cat#: 103116; Clone: 30-F11), Anti-mouse FcεRIα-PE/cy7, (BioLegend; Cat#: 334620; Clone: AER-37 (CRA-1)), Anti-mouse CD117 (Kit)-APC, (BioLegend; Cat#: 161505; Clone: S18020A), and Zombie Aqua™ Fixable Viability Kit, (BioLegend; Cat#: 423101)] was added and incubated for 30 min at 4 °C[39]. The stained cells were analyzed by FACS LSRFORTESSA (BD, USA), and the data were analyzed by Flowjo software.

## Cell culture

Bone marrow-derived MCs (BMMCs) were differentiated from bone marrow in RPMI 1640 media [+fetal bovine serum (FBS, 10%), penicillin (100 U/mL), streptomycin (100 mg/mL), L-glutamine (2 mM), sodium pyruvate (1 mM), HEPES (10 mM) and recombinant cytokines (stem cell factor, 20 ng/mL; IL3, 20 ng/mL)] for 21 days at 37 °C in 5% $CO_2$[84]. Differentiation was monitored by the expression of CD117 (c-kit) and FcεRIα by fluorescence-activated cell sorting (FACS). RBL-2H3 was purchased from ATCC (CRL-2256) and cultured in MEM-EBSS medium (Hyclone, SH30024.01) supplemented with 10% fetal bovine serum at 37°C and in 5%$CO_2$. The authentication of RBL-2H3 can be acquired from ATCC website according to ATCC number.

## IgE-mediated mast cell activation

Mast cell activation was assessed by β-hexosaminidase release in BMMCs[84,85]. In short, BMMCs were inoculated into 96 well plates with $8*10^3$ cells per well and sensitized with 0.1ug/ml IgE (D8406, Sigma-Aldrich) prepared with serum-free medium. After sensitization for 24 h, the culture medium was discarded and LPS (L2630, Sigma-Aldrich) complete culture medium was added. After 6 h, the medium was discarded and 1 ug/ml DNP-HSA (D-5059, Biosearch Technologies) was added for 30 min.

The supernatant of each well was collected, and NP40 Lysis Buffer was added to each well to lyse the cells for 5 min. Lysate (50 ul) and supernatant (50 ul) were each incubated with substrate solution (50 μL) (3 mM 4-Nitrophenyl N-acetyl-β-D-glucosaminide, N9376, Sigma-Aldrich) for 90 min at 37 °C, then 150 uL of $Na_2CO_3$/$NaHCO_3$ (pH = 10.6) were added to terminate the reaction, and the absorbance value was read at 405 nm by Cytation 5 multifunctional enzyme marker (BioTek, USA), and the degranulation ratio of mast cells was obtained by supernatant/(supernatant + lysate) absorbance value.

## Statistical analysis

Data with normal distribution were analyzed by two-tailed *t* test or ANOVA, and results are shown as mean ± s.e.m. Not normally distributed data were analyzed by Wilcoxon test, and results are expressed as median (25% percentile–75% percentile). *P* values < 0.05 were considered statistically significant.

## Reporting summary

Further information on research design is available in the Nature Portfolio Reporting Summary linked to this article.

# Data availability

Data that support the findings of this study are available within the paper and its Supplementary Information. The metagenomic sequencing files of human fecal samples and the 16S rRNA gene sequencing files of fecal samples from experimental mice generated in this study have been deposited in the National Omics Data Encyclopedia (NODE). The metagenomic sequencing files of human fecal samples are available under project OEP002960 (http://www.biosino.org/node/review/

detail/OEV000435?code=K4MN2WY7), the 16S rRNA gene sequencing files of fecal samples from experimental mice before and after antibiotic treatment are under project OEP002997 (http://www.biosino.org/node/review/detail/OEV000436?code=TN2WICKS). The mass spectrometry data generated in this study have been deposited in the China Nucleic Acid RepositoRy Database (CNGBdb, https://db.cngb.org/cnjb/) under accession CNP0005021. The experimental data generated in this study are provided in the Source Data file. Source data are provided with this paper.

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

## Acknowledgements

We thank Dr. Wen Zhou and Dr. Yinghong Zhu from the Institute of oncology, School of basic medicine, Central South University for providing *Klebsiella pneumoniae* and experimental technical support. Thanks to Dr. Ben Lu, Key Laboratory of Sepsis Translational Medicine of Hunan and Department of Critical Care Medicine and Hematology of the 3rd Xiangya Hospital, Central South University for kindly gifting *TLR4* $^{+/+}$ and *TLR4* $^{-/-}$ mice. In addition, this study was supported by fundings provided by the National Natural Science Foundation of China, Grant No. 81974476 (J.L.), 82173424) (J.L.), 82073458 (C.P.) and 81830096 (X.C.), respectively. This work was also supported by the Science and Technology Innovation Program of Hunan Province 2021RC4013 (C.P.) and grant from the Scientific Research Program of FuRong Laboratory (No. 2023SK2103), the Program of Introducing Talents of Discipline to Universities, 111 Project, No. B20017 (X.C.), and the Graduate Students Explore Innovative Programs of Central South University, CX20220338 (L.Z.).

## Author contributions

L.Z., X.J., B.Z. and R.L. contributed equally to this work. J.L. and R.L. collected stool samples and plasma samples of CSU patients.R.L., B.Z., J.L., and M.Ma prepared the draft of the manuscpirt. L.Z., J.L., and M.Ma prepared the revised verion of manuscript. L.Z. did cell experiments and animal experiments, data analysis,data collection, and prepared Figs. 2–4, 6 and Supplementary Figs. 5, 6. B.Z. conducted animal experiments, analyzed experimental data and prepared Figs. 3–7, Supplementary Fig. 4, Supplementary Tables 1–3. X. J. performed metagenomic sequencing analysis and prepared Fig. 1 and Supplementary data Fig. 1. R.L. conducted part of the animal experiments, analyzed 16 S rRNA gene sequencing data. W.S. provided the data management. M.Ma. and M.Mu; contributed to data interpretation. L.X. contributed to bioinformatics support. J.L.and M.Ma formulated the study and provided supervision. J.L., M.Ma, C.P., X.C. and M.Mu, guided the research design. J.L., M.Ma, and C.P. developed and revised the manuscript and contributed to data presentation. All authors reviewed the manuscript.

## Funding

## Competing interests

M.Ma is or recently was a speaker and/or advisor for and/or has received research funding from Astria, Allakos, Alnylam, Amgen, Aralez, ArgenX, AstraZeneca, BioCryst, Blueprint, Celldex, Centogene, CSL Behring, Dyax, FAES, Genentech, GIInnovation, GSK, Innate Pharma, Kalvista, Kyowa Kirin, Leo Pharma, Lilly, Menarini, Moxie, Novartis, Pfizer, Pharming, Pharvaris, Roche, Sanofi/Regeneron, Shire/Takeda, Third Harmonic Bio, UCB, and Uriach. All other authors declare no competing interests.
