## [Peer Review File · Nature Communications]

Gut microbiota facilitate chronic spontaneous urticariaREVIEWER COMMENTS

Reviewer #1 (Remarks to the Author):

In this paper the authors find that the fecal microbiome is altered in patients with chronic spontaneous urticaria (CSU) relative to healthy controls. The altered microbiome is also correlated with lower levels of SCFAs and higher circulating LPS levels. Upon transfer of human microbiota into mice, they can observe evidence of increase gut permeability and mast cell activation. Transfer of *K. pneumoniae* (increased in CSU patients) increase mast cell activation and gut permeability, while transfer of *Roseburia hominis* (or treatment with a SCFA caporate) is attenuate mast cell activation and lower gut permeability.

The basic model proposed by the authors is the standard 'LPS is bad, SCFA are good'. This idea is well articulated. However, it is likely that the real situation is more nuanced. For example, Vetanen et al. (Cell, 2016) have shown that there are different forms of LPS, and while Proteobacteria-derived LPS is a stronger TLR4 stimulant relative to Bacteroides-derived LPS, it also induces LPS tolerance. Thus, it is not obvious that greater abundance of Proteobacteria is necessarily a driver of inflammation.

The transfer of human microbiota into mice is a reasonable experiment, although the mouse has vastly different gastrointestinal physiology relative to human (e.g., it is coprophagic), which makes it difficult to extrapolate.

I suggest that the discussion of the paper should be elevated beyond the simplistic model of LPS vs SCFAs. As noted above, not all LPS is the same and a completely different conclusion from elevated LPS levels is also possible. Similarly, SCFAs constitute just one of several major classes of microbiota-derived compounds that are typically considered in discussions of gut barrier function and inflammation. At least a comment on aryl carbohydrate receptor ligands and secondary bile acids should be made.

The discussion should acknowledge that fecal composition in humans is only a limited window on the gut microbiome. In fact, it is likely that the small bowel is more important. It should mention the difficulty of measuring gut permeability in humans, although human studies of the gut barrier function using different sugar inputs do exist in CSU.

The human cohort should be better described. Autoimmune conditions are common in CSU patients. SLE was an exclusion criterion. What about other autoimmune conditions, e.g., Sjogern's, thyroid problems. Were autoimmune antibodies measured? Did the participants have any gastrointestinal symptoms?

Minor:

Line 104. The referenced paper did not show decreased levels of Bacteroides in serum.

Line 208. Claiming that this is the first study that shows an altered fecal microbiome in CSU contradicts the author's introduction and later discussion, which cites previous papers showing altered fecal microbiome in CSU.

Reviewer #2 (Remarks to the Author):

In this manuscript, Zhou et al. demonstrated an association between specific microbiota and the pathogenesis of chronic spontaneous urticaria (CSU). They showed that fecal microbial transplantation derived from patients with CSU facilitated IgE-mediated MC-driven skin inflammatory responses via increased intestinal permeability and blood LPS accumulation in recipient mice. The presented data are

potentially interesting, however, there are some major concerns to be addressed.

1. The authors concluded that increased and reduced LPS and SCFA levels by altered gut microbiome play key roles in mast cell-driven skin inflammation. The causal relationship between specific microbiota, LPS/SCFA, and skin inflammation remains obscure. TLR4 or GPR deficient mice should be used to address these points.
2. Overall, the evaluation of mast cells is insufficient. The authors only demonstrated the proportion of degranulated mast cells, but the number, phenotype, and the function (IgE production, inflammatory cytokines production, etc.) should be demonstrated. Flowcytometry analysis would be useful.
3. How gut microbiota derived from CSU patients affect the intestinal permeability? Although the authors described the possible mechanism in discussion, they need to show in this murine model. FMT of gut microbiota from CSU patients affect intestinal epithelial tight junction or intestinal inflammation in the recipient mice?
4. It is uncertain why the authors finally focused on *Klebsiella pneumoniae* (Kp)? Although they showed that Kp was positively correlated with CSU disease activity, the specific role in the pathogenesis of CSU has not been fully addressed. The Kp strain from ATCC was used in vivo study, however, the authors need to show whether the specific strain isolated from CFU patients has unique characteristics that make it particularly inflammatory compared to other Kp strains. I think this point is critical.
5. In experiments with antibiotics to eradicate Kp (Figure 4F), quantification of Kp and the overall composition of bacterial flora need to be shown.

Reviewer #3 (Remarks to the Author):

The study by Zhou, Jian and Liu investigates the relationship between chronic spontaneous urticaria (CSU) and gut dysbiosis using metagenomics sequencing and short-chain fatty acids (SCFAs) metabolomics. The researchers showed that CSU gut microbiota has low diversity and SCFA production but high levels of *Klebsiella pneumoniae*, a conditional pathogenic bacteria negatively correlated with blood SCFA levels and linked to high disease activity. Blood LP levels were elevated, linked to rapid disease relapse and high gut levels of conditionally pathogenic bacteria. FMT of CSU microbiome and *Klebsiella* facilitated skin inflammatory responses and increased intestinal permeability and blood LPS accumulation in recipient mice. In contrast, transplantation of *Roseburia* and caproic acid administration protected recipient mice from MC-driven skin inflammation. The study suggests that gut microbiome alterations in CSU may reduce SCFA levels, increase LPS levels, and facilitate MC-driven skin inflammation.

The study appears to have a strong design for both human and mouse components and is well-documented, but some points require clarification.

A crucial aspect is the statistical analysis of the microbiota, and it is unclear whether the authors considered the patients' clinical metadata. It is important to utilize a comprehensive pipeline to determine multivariable associations between phenotypes, covariates, and microbial meta'omic features efficiently.

Additionally, the study lacks information on the adj-p values for microbiota and metabolomic analysis. It would be interesting to see how many of the presented data remain significant after correcting for multiple comparisons using p.adjust. While the study's message may not change, and the most critical taxa will likely remain significant, the authors should correct all the statistics presented in their manuscript. I will suggest using a pipeline such as MaAsLin2, CoDaSeq or ALDEx2.

The authors classified the bacteria into two categories: SCFA producers and pathobionts, which appear to be accurate based on the taxa identified in their study. However, it would be beneficial if they further clarified how they categorized these bacteria.

What are the keystone taxa in each network analysis performed? Does "the key hub species" mean keystone taxa? If so, can the authors calculate the Between Centrality Score to identify the most influential taxa in each group?

Figure 2D and E: what are the level in control (non-treatment at all) or just antibiotic-treated but no FMT performed groups of mice? It will be important to see the basal level.

Figure 3: Could the authors provide more information on the composition of the control group in their study? It is stated that a control group is an antibiotic-treated group, but it is unclear if there was a group of totally normal colonized SPF mice used as a control. Including such a control group would provide valuable information on the effects of antibiotics on the phenotypes observed in the study, and whether they can be reversed to a basal state. Therefore, it would be beneficial if the authors could provide clarification on whether a group of completely untreated SPF mice was used as a control and, if not, the rationale for selecting an antibiotic-treated group as the control.

While reviewing the study, I attempted to access the xlsx files containing the raw data and statistical results provided by the authors. However, I found the files to be poorly labeled, with names such as "420201_0_extended_data_4098726_rry2c9.xlsx" or "420201_0_source_data_4098731_rr62c9.xlsx". Although the authors' effort to provide all the raw data is appreciated, it was time-consuming to navigate between the manuscript tracking system and the data files to identify the correct table number. While this is a minor issue, I would suggest that the authors deposit their files with more informative annotations in the future to make them more accessible to readers.

Furthermore, a more significant concern is the absence of legends for the tables. I was unable to locate the legends for the tables, and in some cases, some xlsx files contained multiple tabs or several panels within the same tab. While this may be the format required by the journal, I found it to be an unusual way of reporting data. Providing clear legends and organizing the data in a more intuitive manner would improve the readability and usability of the raw data for readers.

Questions for the methodology section:

- Can you provide more information about the collection tubes that contain DNA preservation solution? The methodology section does not specify the exact type or brand of tubes used. It would be helpful to know the specific details of the tubes to better understand the preservation method and potential effects on the microbiota profiles.

- Have you ever sequenced the cotton swab itself to determine if it contributes to the microbiota profile? Despite the authors' efforts to exclude extraneous DNA, it is important to consider potential contamination from the swab. It would be interesting to know if the swab has its own distinct microbial signature and to what extent it may affect the results.

- Line 400: 100um screen. What is this exactly?

- Macrogenomic sequencing. I am not aware of this terminology. Would it be more appropriate to refer to this as amplicon sequencing or another more common term?

- The methodology section states that the authors sequenced the v3-v4 region using amplicon sequencing, but then fragmented the amplicon. Could you clarify the details of the primers used and the size of the amplicon? If the amplicon was already shorter than 600bp, it would be helpful to explain why it was necessary to fragment it further.

Line 132: Supplementary Figure 1. I couldn't find this figure.

Line 146: I believe that the figure referred to in this paragraph is incorrectly labeled as Extended Data Figure 4a.

Line 176 - 181: This part of the study shows the longitudinal analysis however, it is a bit confusing since it is not well described the terms in the text or legend. What do you mean by status: high risk or low risk? Can you expand this results' interpretation so that we can understand better? What do you mean by survival probability? Does anyone die from this disease? Sorry for my ignorance.

**Point-by-point response to the Reviewers**

Date: 24-Aug-2023

Manuscript Number: NCOMMS-23-12292-T

Title of Article: Gut microbiota facilitate chronic spontaneous urticaria

Names of the Corresponding Authors: Jie Li, Marcus Maurer

Email Addresses of the Corresponding Authors: xyljie@csu.edu.cn,

marcus.maurer@charite.de

Dear reviewers,

Thank you for your comments on our manuscript "Gut microbiota facilitate chronic
spontaneous urticaria" . Your feedback, input, and suggestions were very helpful for
revising and improving our paper. We carefully studied your comments and performed
additional experiments and made corrections in response to them as detailed below and
highlighted in yellow in the revised version of our manuscript.

Major changes and additions to the revised manuscript:

- 1. Added results of additional experiments evaluating mast cells
- 2. Added results of additional experiments with TLR4-deficient mice and mast cells
- 3. Added results of additional experiments with GPR41-deficient mice
- 4. Added results of additional FMT experiments that assessed recipient mice for changes
in intestinal permeability and expression of barrier markers
- 5. Added results of FMT experiments without prior antibiotic treatment
- 6. Included discussion of other potential mechanisms

REVIEWER COMMENTS

Reviewer #1 (Remarks to the Author):

In this paper the authors find that the fecal microbiome is altered in patients with chronic
spontaneous urticaria (CSU) relative to healthy controls. The altered microbiome is also

correlated with lower levels of SCFAs and higher circulating LPS levels. Upon transfer of
human microbiota into mice, they can observe evidence of increase gut permeability and
mast cell activation. Transfer of *K. pneumoniae* (increased in CSU patients) increase mast
cell activation and gut permeability, while transfer of *Roseburia hominis* (or treatment
with a SCFA caporate) is attenuate mast cell activation and lower gut permeability.

**Overall response:** We sincerely thank the reviewer for the thorough review of our work
and the insightful and helpful comments, all of which have been addressed, as detailed
below.

(1) The basic model proposed by the authors is the standard 'LPS is bad, SCFA are
good'. This idea is well articulated. However, it is likely that the real situation is more
nuanced. For example, Vetanen et al. (Cell, 2016) have shown that there are different
forms of LPS, and while Proteobacteria-derived LPS is a stronger TLR4 stimulant relative
to Bacteroides-derived LPS, it also induces LPS tolerance. Thus, it is not obvious that
greater abundance of Proteobacteria is necessarily a driver of inflammation.

**Response:** Thanks for raising this point, it is well taken. Indeed, the role, relevance, and
effects of LPS in the regulation of immunity are complex. We changed and added new
wording, in the revised version of our manuscript, to reflect this. The added wording
includes discussion of the Vetanen et al. paper (added as reference 64 in our revised
manuscript) as well as evidence pointing to the various effects of LPS from different
bacterial sources including the induction of LPS tolerance and the enhancing effects of
low-dose LPS exposure on allergic inflammation in OVA-sensitized mice, with higher
levels of Th2 cytokine production (Doreswamy and Peden 2011; Kumar and Adhikari
2017). This is important, as CSU is a low-grade chronic inflammatory condition. Unlike
bacterial infections, where large amounts of endotoxin are released into the circulation,
plasma levels of LPS in CSU result from enhanced intestinal permeability, a long-term
and slow process that is expected to lead to continued but moderately elevated blood
levels, as supported by our results.

The changed and added wording included in the revised version of our manuscript reads
as follows: “LPS, via TLRs (TLR4), can activate MCs to produce various inflammatory
mediators such as TNF- α and IL-13^{61,62} and facilitates and enhances IgE-mediated MC
degranulation⁶³, as confirmed by our results. It has to be kept in mind, though, that the
regulation of immunity by LPS is a complex process, with differences of Bacteroides LPS
as compared to Escherichia coli LPS related to immunogenicity and endotoxin tolerance⁶⁴.
Furthermore, Sudhir et al. reported that low-dose LPS exposure enhanced allergic airway
inflammation in OVA-sensitized mice, with higher levels of Th2 cytokine production and
reduced IFN- γ production, whereas high-dose LPS contributed to the maintenance of
allergic immune tolerance by elevating Tregs⁶⁵. Further studies are needed to characterize,
in detail, the mechanisms that lead to increased intestinal permeability in CSU and how
the translocation of LPS to the bloodstream affects CSU diseases activity and duration.”

(2)The transfer of human microbiota into mice is a reasonable experiment, although the
mouse has vastly different gastrointestinal physiology relative to human (e.g., it is
coprophagic), which makes it difficult to extrapolate.

**Response:** Thanks for this helpful comment. We agree that mice and men differ greatly
in gastrointestinal physiology. We used this approach, fecal microbial transfer to mice, as
a model. While this model is well established and widely used to study the role of human
gut microbes in disease, we acknowledge its limitations. In the revised version of our
manuscript, we extend on this, discussing the results of previous studies that used this
model, calling for caution in the extrapolation of results to the human system, and
encouraging further studies including fecal transplantation studies in humans.

The wording added to the revised version of our manuscript reads as follows: “Fecal
microbial transfer to mice is a well-established and widely used model to study the
pathogenic role of human gut microbes, for example in Alzheimer's disease (Chen, et al.
2022), major depressive disorder, (Zhang, et al. 2022), and chronic kidney disease (Wang,
et al. 2020). However, mice and men differ greatly in gastrointestinal physiology and

extrapolating the results of mouse microbial transfer studies to the human system must
be done with caution. Further studies including fecal transplantation studies in humans
are needed to confirm and better characterize the role of gut microbiota in chronic
spontaneous urticaria.”

(3) I suggest that the discussion of the paper should be elevated beyond the simplistic
model of LPS vs SCFAs. As noted above, not all LPS is the same and a completely different
conclusion from elevated LPS levels is also possible. Similarly, SCFAs constitute just one
of several major classes of microbiota-derived compounds that are typically considered
in discussions of gut barrier function and inflammation. At least a comment on aryl
carbohydrate receptor ligands and secondary bile acids should be made.

**Response:** The reviewer is correct, thanks for the constructive comments. We revised the
discussion of our paper and added wording on the potential role of microbiota-derived
compounds other than LPS and SCFA including aryl carbohydrate receptor ligands: “As
for limitations, our study did not assess the levels and effects of metabolic bacterial
products other than LPS and SCFA, but there are many additional ones that may also play
a role in CSU, including ligands of the Aryl hydrocarbon receptor (AhR), an important
modulator of immune and inflammatory diseases⁶⁸. Gut bacteria may also affect CSU by
modifying primary bile acids and generating secondary bile acids (SBAs) such as
deoxycholic acid, lithocholic acid and ursodeoxycholic acid, which are implicated in both
innate and adaptive immune responses⁶⁹. In autoimmune diseases, such as type 1
diabetes, there is a notable increase in the presence of bacteria that produce SBA⁷⁰, and
Elena et al. reported a higher abundance of bile acids (specifically taurocholate) in
children suffering from asthma compared to the control group. The elevated levels of bile
acids may be associated with the phenotypic expression and pathogenesis of asthma⁷¹.
These and other potential mechanisms of microbiome effects on CSU should be
investigated in future studies.”

(4) The discussion should acknowledge that fecal composition in humans is only a limited
window on the gut microbiome. In fact, it is likely that the small bowel is more important.

It should mention the difficulty of measuring gut permeability in humans, although
human studies of the gut barrier function using different sugar inputs do exist in CSU.

**Response :** We thank the reviewer for this helpful suggestion. We revised the discussion
accordingly. The wording we added reads as follows: "Human fecal composition is only
partially representative of the composition of the intestinal microbiome, and most
nutrients are absorbed through the small intestine in the physiological state."

In our study, we used the FITC-dextran assay to assess intestinal permeability, a method
that is now widely used (Dawson, et al. 2009; Sorribas, et al. 2019; Woting and Blaut 2018).
In humans, performing intestinal permeability assays usually requires obtaining biopsies
of intestinal tissue, which is an invasive test and often difficult to implement in practical
studies. Moreover, an *in vivo* triple glucose test has been used to measure gastrointestinal
permeability in chronic spontaneous urticaria (Buhner, et al. 2004).

(5) The human cohort should be better described. Autoimmune conditions are common
in CSU patients. SLE was an exclusion criterion. What about other autoimmune conditions,
e.g., Sjogern' s, thyroid problems. Were autoimmune antibodies measured? Did the
participants have any gastrointestinal symptoms?

**Response :** Thank you for your comment. We obtained detailed data on the medical
history, comorbidities, and clinical manifestation of all our patients. We also performed
laboratory tests such as routine blood analyses, autoantibodies (e.g., anti-thyroid
autoantibodies, SLE-related antibodies), sedimentation, CRP, and thyroid hormone levels.
Patients with comorbid autoimmune disease and/or gastrointestinal symptoms were
excluded from this study. We now provide a more detailed description of the inclusion
and exclusion criteria applied, in the material and methods section of the revised version
of our manuscript: "The inclusion criteria for patients with CSU were: 1) Aged 18 to 65;
2) No oral or topical antihistamines within one month before sample collection; 3) No
antibiotics, prebiotics, probiotics, glucocorticoids, omalizumab and other drugs within
three months before sample collection; 4) No consumption of yogurt, pickles and other
fermented foods within three days before sample collection; 6) Living in Changsha city

for more than 1 year before sample collection. The exclusion criteria were: 1) Suffering
from other subtypes of urticaria, such as symptomatic dermographism and acute urticaria,
or allergic diseases; 2) With comorbid autoimmune disease (such as systemic lupus
erythematosus, Sjogren' s syndrome, thyroid problems, diabetes) and/or
gastrointestinal symptoms; 3) Failure to collect fecal samples as required; 4) Lactation or
pregnancy."

Minor:

(6) Line 104. The referenced paper did not show decreased levels of Bacteroides in serum.

**Response:** We thank the reviewer for catching this mistake. We corrected it in the revised
version of our paper.

(7) Line 208. Claiming that this is the first study that shows an altered fecal microbiome
in CSU contradicts the author' s introduction and later discussion, which cites previous
papers showing altered fecal microbiome in CSU.

**Response:** Thank you for this helpful feedback. We changed this in the revised version of
our paper.

Reviewer #2 (Remarks to the Author):

In this manuscript, Zhou et al. demonstrated an association between specific microbiota
and the pathogenesis of chronic spontaneous urticaria (CSU). They showed that fecal
microbial transplantation derived from patients with CSU facilitated IgE-mediated MC-
driven skin inflammatory responses via increased intestinal permeability and blood LPS

accumulation in recipient mice. The presented data are potentially interesting, however,
there are some major concerns to be addressed.

**Overall response:** We thank the reviewer for their thorough review of our report and the
insightful comments and suggestions, all of which have been addressed, as detailed
below.

(1) The authors concluded that increased and reduced LPS and SCFA levels by altered gut
microbiome play key roles in mast cell-driven skin inflammation. The causal relationship
between specific microbiota, LPS/SCFA, and skin inflammation remains obscure. TLR4 or
GPR deficient mice should be used to address these points.

**Response:** We thank the reviewer for this helpful suggestion. We performed several
additional *in vivo* and *in vitro* experiments, and the results further strengthen the
conclusions of our study.

First, we gavaged TLR4 KO mice with LPS for 1 week and then subjected them to passive
cutaneous anaphylaxis. As compared to wild type control mice, the promotion of passive
cutaneous anaphylaxis by LPS was markedly and significantly attenuated in TLR4 KO mice.
TLR4 KO mice had significantly lower mast cell infiltration, mRNA expression, and
tissue/plasma levels of type 2 inflammatory cytokines and histamine. These results
suggest that the promotion of allergic skin inflammatory responses by LPS are, at least in
part, mediated by TLR4. The findings from these additional experiments are shown in
Figure 6a-g of the revised version of our manuscript.

To better characterize the role and relevance of TLR4 in LPS-promoted mast cell
activation, we performed functional *in vitro* assays with mast cells isolated from TLR4 KO
or wild-type control mice. As compared to mast cells from wild type mice, TLR4-deficient
mast cells showed lower levels of activation and cell mobility after incubation with LPS
and IgE-mediated activation, shown in Figure 6h of our revised paper.

Fig. 6:

In addition, we verified that the application of caproate attenuates IgE-induced mast
 cell degranulation, inflammatory cytokines and histamine release (Figure 4 of the revised
 version of our manuscript).

Fig. 4:

 Knock down of GPR41 failed to block this effect of caproate on mast cell activation
 (Supplementary Fig. 5 of the revised version of our manuscript). This indicates that
 caproate suppresses mast cell activation independent of GPR41, which is in line with the
 results of previous studies that showed that propionate and butyrate inhibiting IgE- and
 non-IgE-mediated degranulation of human or mouse mast cells through histone
 deacetylases, but not depending on SCFA receptors, such as GPR41, GPR43 or PPAR
 (Folkerts, et al. 2020).

Supplementary Fig. 5

(2) Overall, the evaluation of mast cells is insufficient. The authors only demonstrated the
 proportion of degranulated mast cells, but the number, phenotype, and the function (IgE
 production, inflammatory cytokines production, etc.) should be demonstrated.
 Flowcytometry analysis would be useful.

**Response:** We thank the reviewer for this helpful comment. In our additional experiments,
 we analyzed the number of skin mast cells in LPS-treated TLR4 KO and wild type mice
 subjected to passive cutaneous allergic reactions using flow cytometry. The results are
 shown in Figures 6a-d of the revised version of our manuscript. We also examined the
 levels of histamine release from mast cells and the effects of LPS/SCFAs on the migratory
 capacity of mast cells. The findings from these experiments are shown in Figures 6h of
 the revised version of our manuscript.

Fig. 6:

(3) How gut microbiota derived from CSU patients affect the intestinal permeability?

Although the authors described the possible mechanism in discussion, they need to show

in this murine model. FMT of gut microbiota from CSU patients affect intestinal epithelial

tight junction or intestinal inflammation in the recipient mice?

**Response:** We thank the reviewer for this interesting and important question. To answer

it, we repeated the FMT experiments and assessed recipient mice for changes in intestinal

permeability. FMT with intestinal flora of CSU patients significantly increased the intestinal

permeability of recipient mice as compared to FMT with intestinal flora of HC.

Furthermore, to better characterize the effect of CSU FMT on mucosal integrity, we

assessed the expression of epithelial tight junction molecules in the colon of recipient

mice (Figure 2e and f of the revised version of our manuscript). CSU FMT down-regulated
 mRNA expression and protein levels of barrier function markers including ZO-1, MUC-2,
 occludin, TJP2, and CGN (Figure 2g of the revised version of our manuscript). In addition,
 CSU FMT up-regulated skin mRNA expression of IL-4, IL-13, TNF- α , and IL-13 mRNA,
 skin or plasma levels of IL-4, IL-13, and histamine, as well as plasma LPS levels (Figure
 2h-k of the revised version of our manuscript). Taken together, these additional findings
 support that the adoptive transfer of gut microbiota from CSU patients affects intestinal
 permeability and promotes inflammation in recipient mice.

Fig. 2:

(4) It is uncertain why the authors finally focused on *Klebsiella pneumoniae* (Kp)?
 Although they showed that Kp was positively correlated with CSU disease activity, the
 specific role in the pathogenesis of CSU has not been fully addressed. The Kp strain from
 ATCC was used in vivo study, however, the authors need to show whether the specific

strain isolated from CSU patients has unique characteristics that make it particularly
 inflammatory compared to other Kp strains. I think this point is critical.

**Response:** Thank you for raising this important point, it is well taken. In this study,
 *Klebsiella pneumoniae* (KpN) was one of several species enriched in CSU (shown in Figure
 1b of the revised version of our manuscript) and positively correlated with CSU disease
 activity (shown in Figure 1f of the revised version of our manuscript).

Fig.1b

Fig. 1f

To explore the relationship between KpN strains enriched in CSU and the KpN strain we
 used in the mouse experiments, we conducted a bioinformatics pipeline. After performing
 quality control and removing human genome sequences by using Kneaddata (version
 0.12.0), five metagenomics sequenced files with the most abundant KpN were selected
 and analyzed. The relative abundance evaluated by HUMAnN2 as described in the
 materials and methods section of the revised version of our manuscript. We performed
 taxa annotation by using Kraken2 (version 2.1.3), and sequencing reads (fastq format)
 from KpN strains were extracted by using function "extract_kraken_reads.py" . Then,
 extracted reads from each sample were separately used for sequence assembly by using

MEGHIT (version 1.2.9). We singled out 15 sequences (>3000bp, fasta format) for
 subsequent alignment (2~4 sequences from the final contigs of each sample). These 15
 contigs were treated as unknown sequences to separately perform BLASTn in the NCBI
 database. We found that a variety of KPn strains were identified with >98% coverage and
 0.0 E-value (Table 1), indicated that those KPn strains are present in the samples of CSU
 patients. We then analysed the KPn strain (ATCC 10031) used in our study. To this end,
 we downloaded the referenced genome sequence from the ATCC website and we
 conducted whole-genome sequencing (Novogene, Beijing, China) for this KPn strain.
 After sequence blasting, we found that the similarity between the two exceeded 99% using
 BLAST (version 2.12.0): Score (bits)=4.233e+06; E value=0.0; Identities=2292282/2292367
 (99%); Gaps=18/2292367 (0%), confirming that the experimental KPn strain we used was
 indeed ATCC 10031. Then, we used the 15 contigs from CSU samples for sequence
 alignment by using BLAST. Consistently, we observed that those contigs showed >97%
 similarity and 0.0 E-value with ATCC 10031. Taken together, our results support that the
 effects of the KPn strain we used for our experiments reflect those of the KPn strains in
 the gut of CSU patients (Table 2).

Table 1. The top 5 alignment results of the 15 contigs derived from CSU samples in NCBI database.

Description	Taxid	Max Score	Total Score	Query cover	E Value	Per. Ident	Acc. Len	Accession
Job Title: k141_3611 AF20000182 len=3219								
Klebsiella pneumoniae strain A16KP0127	573	5675	5675	99%	0.0	98.51	5352208	CP052565.1
Klebsiella pneumoniae strain F17KP0054	573	5670	5670	99%	0.0	98.42	5357257	CP052136.1
Klebsiella pneumoniae strain Bckp101	573	5670	5670	99%	0.0	98.42	5327744	CP050840.1
Klebsiella pneumoniae strain Bckp212	573	5670	5670	99%	0.0	98.42	5204371	CP050826.1
Klebsiella pneumoniae strain 205880	573	5670	5670	99%	0.0	98.42	5212950	CP030302.1
Job Title: k141_4414 AF20000182 len=3371								
Klebsiella pneumoniae isolate 104 genome assembly	573	6131	6386	100%	0.0	99.5	5334698	OW848825.1
Klebsiella pneumoniae strain B16KP0048	573	6115	6375	100%	0.0	99.41	5315357	CP052716.1
Klebsiella pneumoniae strain B16KP0078	573	6115	6375	100%	0.0	99.41	5350660	CP052706.1
Klebsiella pneumoniae strain B17KP0020	573	6115	6375	100%	0.0	99.41	5308267	CP052503.1
Klebsiella pneumoniae strain B16KP0198	573	6115	6375	100%	0.0	99.41	5309291	CP052520.1
Job Title:k141_1437 AF20000182 len=3104								
Klebsiella pneumoniae strain B17KP0020	573	5594	5594	100%	0.0	99.19	5308267	CP052503.1
Klebsiella pneumoniae strain B16KP0198	573	5594	5594	100%	0.0	99.19	5309291	CP052520.1
Klebsiella pneumoniae strain D16KP0042	573	5594	5594	100%	0.0	99.19	5345365	CP052372.1
Klebsiella pneumoniae strain B16KP0183	573	5594	5594	100%	0.0	99.19	5307940	CP052522.1
Klebsiella pneumoniae strain E16KP0210	573	5594	5594	100%	0.0	99.19	5323430	CP052295.1
Job Title:k141_975 AF20000201 len=3959								

Klebsiella pneumoniae strain SB617	573	7243	7243	99%	0.0	99.7	5301905	CP084825.1
Klebsiella pneumoniae strain RHB30-C05	573	7243	7243	99%	0.0	99.7	5146480	CP057313.1
Klebsiella pneumoniae strain 1050	573	7239	7239	100%	0.0	99.67	5338202	CP023416.1
Klebsiella pneumoniae strain 59062CZ	573	7239	7239	100%	0.0	99.67	5350752	CP085729.1
Klebsiella pneumoniae strain 2017-45-36	573	7238	7238	99%	0.0	99.67	5312404	CP109703.1
Job Title:k141_755 AF20000201 len=4041								
Klebsiella pneumoniae strain KP2	573	5321	7941	100%	0.0	97.86	5349282	CP041946.1
Klebsiella pneumoniae subsp. pneumoniae strain BK13048	72407	5321	7941	100%	0.0	97.86	5213293	CP045015.1
Klebsiella pneumoniae strain 18-2374	573	5321	7941	100%	0.0	97.74	5253969	CP041927.1
Klebsiella pneumoniae strain PIMB15ND2KP27	573	5321	7941	100%	0.0	97.86	5247824	CP041639.1
Klebsiella pneumoniae strain AR_0075	573	5321	7941	100%	0.0	97.74	5312321	CP032185.1
Job Title:k141_1754 AF20000204 len=9960								
Klebsiella pneumoniae strain M297-1	573	18377	19969	100%	0.0	99.97	5301757	CP051490.1
Klebsiella pneumoniae strain 205880	573	18371	19963	100%	0.0	99.96	5212950	CP030302.1
Klebsiella pneumoniae subsp. pneumoniae strain KP67	72407	18371	19963	100%	0.0	99.96	5287345	CP101560.1
Klebsiella pneumoniae strain fekpn2511	573	18371	19899	100%	0.0	99.96	5220907	CP068972.1
Klebsiella pneumoniae strain KPN55602	573	18000	19312	100%	0.0	99.29	5127934	CP042977.1
Job Title:k141_244 AF20000204 len=6703								
Klebsiella pneumoniae strain 205880	573	12368	12368	100%	0.0	99.97	5212950	CP030302.1
Klebsiella pneumoniae subsp. pneumoniae strain KP67	72407	12368	12368	100%	0.0	99.97	5287345	CP101560.1
Klebsiella pneumoniae strain fekpn2511	573	12368	12368	100%	0.0	99.97	5220907	CP068972.1
Klebsiella pneumoniae strain M297-1	573	12362	12362	100%	0.0	99.96	5301757	CP051490.1
Klebsiella pneumoniae subsp. pneumoniae strain DEUKp4822	72407	12279	12279	100%	0.0	99.73	5127588	CP113838.1
Job Title:k141_2334 AF20000204 len=10245								
Klebsiella pneumoniae strain 205880	573	18896	18896	100%	0.0	99.96	5212950	CP030302.1
Klebsiella pneumoniae strain DT12	573	18896	18896	100%	0.0	99.96	5210756	CP019079.1
Klebsiella pneumoniae strain DT1	573	18896	18896	100%	0.0	99.96	5166357	CP019077.1
Klebsiella pneumoniae subsp. pneumoniae strain TGH10 chromosom	72407	18896	18896	100%	0.0	99.96	5355459	CP012744.1
Klebsiella pneumoniae subsp. pneumoniae strain TGH8	72407	18896	18896	100%	0.0	99.96	5439720	CP012743.1
Job Title:k141_2361 AF20000204 len=7275								
Klebsiella pneumoniae strain 205880	573	12390	13431	100%	0.0	99.99	5212950	CP030302.1
Klebsiella pneumoniae subsp. pneumoniae strain KP67	72407	12390	13431	100%	0.0	99.99	5287345	CP101560.1
Klebsiella pneumoniae strain M297-1	573	12390	13431	100%	0.0	99.99	5301757	CP051490.1
Klebsiella pneumoniae strain fekpn2511	573	12390	13431	100%	0.0	99.99	5220907	CP068972.1
Klebsiella pneumoniae strain 433	573	12318	13359	100%	0.0	99.76	5218654	CP103621.1
Job Title:k141_2716 AF20000206 len=3757								
Klebsiella pneumoniae strain KPN1343	573	6850	7739	99%	0.0	99.6	5276587	CP033900.1
Klebsiella pneumoniae strain AR_0097	573	6850	6850	100%	0.0	99.57	5309987	CP032200.1
Klebsiella pneumoniae strain R46	573	6850	6850	100%	0.0	99.57	5117042	CP035777.1
Klebsiella pneumoniae strain 2-1	573	6850	6850	100%	0.0	99.57	5230042	CP031562.1
Klebsiella pneumoniae strain F10(AN)	573	6850	6850	100%	0.0	99.57	5272577	CP026153.1
Job Title:k141_5411 AF20000206 len=4036								
Klebsiella pneumoniae strain E16KP0287	573	7348	7348	100%	0.0	99.53	5278060	CP052265.1
Klebsiella pneumoniae strain D17KP0032	573	7348	7348	100%	0.0	99.53	5325688	CP052328.1
Klebsiella pneumoniae strain KPN1343	573	7348	7348	100%	0.0	99.53	5276587	CP033900.1
Klebsiella pneumoniae strain AR_0097	573	7348	7348	100%	0.0	99.53	5309987	CP032200.1

Klebsiella pneumoniae strain 2-1	573	7348	7348	100%	0.0	99.53	5230042	CP031562.1
Job Title:k141_2928 AF20000206 len=3833								
Klebsiella pneumoniae strain D17KP0032	573	7029	7029	100%	0.0	99.77	5325688	CP052328.1
Klebsiella pneumoniae strain KPNIH39	573	7029	7029	100%	0.0	99.77	5351509	CP014762.1
Klebsiella pneumoniae strain kpn-241	573	7029	7029	100%	0.0	99.77	5299480	CP053666.1
Klebsiella pneumoniae strain KSB1_6G	573	7029	7029	100%	0.0	99.77	5333979	CP110545.1
Klebsiella pneumoniae strain KP5076	573	7029	7029	100%	0.0	99.77	5280348	CP106654.1
Job Title:k141_2823 AF20000208 len=4176								
Klebsiella pneumoniae strain B17KP0020	573	6911	7669	100%	0.0	99.6	5308267	CP052503.1
Klebsiella pneumoniae strain B16KP0198	573	6911	7669	100%	0.0	99.6	5309291	CP052520.1
Klebsiella pneumoniae strain D16KP0042	573	6911	7669	100%	0.0	99.6	5345365	CP052372.1
Klebsiella pneumoniae strain B16KP0183	573	6911	7669	100%	0.0	99.6	5307940	CP052522.1
Klebsiella pneumoniae strain Bckp091	573	6911	7652	100%	0.0	99.6	5187296	CP050822.1
Job Title:k141_2045 AF20000208 len=3914								
Klebsiella pneumoniae strain Colony283	573	7206	7206	100%	0.0	99.9	5540574	CP078762.1
Klebsiella pneumoniae strain MGH83	573	7206	7206	100%	0.0	99.9	5343186	CP073054.2
Klebsiella pneumoniae strain C2	573	7184	7184	100%	0.0	99.8	5243002	CP042520.1
Klebsiella pneumoniae strain A16KP0119	573	7179	7179	100%	0.0	99.77	5258127	CP052569.1
Klebsiella pneumoniae strain Kp8701	573	7179	7179	100%	0.0	99.77	5337408	CP049604.1
Job Title:k141_2306 AF20000208 len=3889								
Klebsiella pneumoniae strain TK421	573	7071	7071	100%	0.0	99.49	5277246	CP045694.1
Klebsiella pneumoniae strain KCJ3K307	573	7071	7071	100%	0.0	99.49	5126009	CP054400.1
Klebsiella pneumoniae strain KCJ3K292	573	7071	7071	100%	0.0	99.49	5126008	CP054403.1
Klebsiella pneumoniae strain KCJ3K293	573	7071	7071	100%	0.0	99.49	5125981	CP054406.1
Klebsiella pneumoniae strain 090515	573	7071	7303	100%	0.0	99.49	5060265	CP073287.1

Table 2. The alignment results of the 15 contigs with genome sequence of ATCC 10031.

Contigs ^o	Score (Bites) ^o	E value ^o	Identities ^o	Gaps ^o
k141_3611 AF20000182 len=3219 ^o	5642 ^o	0.0 ^o	3173/3229 (98%) ^o	12/3229 (0%) ^o
k141_4414 AF20000182 len=3371 ^o	6104 ^o	0.0 ^o	3349/3371 (99%) ^o	0/3371 (0%) ^o
k141_1437 AF20000182 len=3104 ^o	5583 ^o	0.0 ^o	3077/3104 (99%) ^o	0/3104 (0%) ^o
k141_2823 AF20000208 len=4176 ^o	6828 ^o	0.0 ^o	3757/3787 (99%) ^o	0/3787 (0%) ^o
k141_2045 AF20000208 len=3914 ^o	7118 ^o	0.0 ^o	3894/3914 (99%) ^o	0/3914 (0%) ^o
k141_2306 AF20000208 len=3889 ^o	5230 ^o	0.0 ^o	2894/2925 (99%) ^o	0/2925 (0%) ^o
k141_1754 AF20000204 len=9960 ^o	18000 ^o	0.0 ^o	9889/9960 (99%) ^o	0/9960 (0%) ^o
k141_2334 AF20000204 len=10245 ^o	18580 ^o	0.0 ^o	10184/10245 (99%) ^o	1/10245 (0%) ^o
k141_2361 AF20000204 len=7275 ^o	6628 ^o	0.0 ^o	3700/3752 (99%) ^o	14/3752 (0%) ^o
k141_244 AF20000204 len=6703 ^o	12168 ^o	0.0 ^o	6665/6703 (99%) ^o	0/6703 (0%) ^o
k141_2716 AF20000206 len=3757 ^o	6756 ^o	0.0 ^o	3724/3757 (99%) ^o	0/3757 (0%) ^o
k141_2928 AF20000206 len=3833 ^o	6951 ^o	0.0 ^o	3810/3833 (99%) ^o	0/3833 (0%) ^o
k141_5411 AF20000206 len=4036 ^o	7321 ^o	0.0 ^o	4012/4036 (99%) ^o	0/4036 (0%) ^o
k141_755 AF20000201 len=4041 ^o	5243 ^o	0.0 ^o	3004/3084 (97%) ^o	10/3084 (0%) ^o
k141_975 AF20000201 len=3959 ^o	7182 ^o	0.0 ^o	3935/3958 (99%) ^o	0/3958 (0%) ^o

(5) In experiments with antibiotics to eradicate Kp (Figure 4F), quantification of Kp and
the overall composition of bacterial flora need to be shown.

**Response:** We thank the reviewer for this helpful suggestion. We analysed mouse feces

from this experiment by real-time PCR to explore the effect of cefoperazone sulbactam
 treatment on *Klebsiella pneumoniae* and other bacteria. We found that the relative
 abundance of KPn and several other conditionally pathogenic bacteria (i.e. *Escherichia*
 *coli*, *Ruminococcus gnavus*, *Bacteroides stercoris*, and *Veillonella parvula*) was
 significantly reduced after administration of cefoperazone sulbactam (shown in
 Supplementary Fig. 6d of the revised version of our manuscript). In contrast, the
 abundance of *Roseburia hominis* was up-regulated after treatment with cefoperazone
 sulbactam (shown in Supplementary Fig. 6d of the revised version of our manuscript).

Supplementary Fig. 6

Reviewer #3 (Remarks to the Author):

The study by Zhou, Jian and Liu investigates the relationship between chronic
 spontaneous urticaria (CSU) and gut dysbiosis using metagenomics sequencing and
 short-chain fatty acids (SCFAs) metabolomics. The researchers showed that CSU gut
 microbiota has low diversity and SCFA production but high levels of *Klebsiella pneumoniae*,
 a conditional pathogenic bacteria negatively correlated with blood SCFA levels and linked
 to high disease activity. Blood LP levels were elevated, linked to rapid disease relapse and
 high gut levels of conditionally pathogenic bacteria. FMT of CSU microbiome and
 *Klebsiella* facilitated skin inflammatory responses and increased intestinal permeability
 and blood LPS accumulation in recipient mice. In contrast, transplantation of *Roseburia*
 and caproic acid administration protected recipient mice from MC-driven skin
 inflammation. The study suggests that gut microbiome alterations in CSU may reduce
 SCFA levels, increase LPS levels, and facilitate MC-driven skin inflammation.

The study appears to have a strong design for both human and mouse components and
is well-documented, but some points require clarification.

**Overall response:** We thank the reviewer for the thorough review and positive evaluation
of our work. We appreciate the insightful and helpful comments and suggestions, all of
which have been addressed, as detailed below.

(1) A crucial aspect is the statistical analysis of the microbiota, and it is unclear whether
the authors considered the patients' clinical metadata. It is important to utilize a
comprehensive pipeline to determine multivariable associations between phenotypes,
covariates, and microbial meta'omic features efficiently.

**Response:** We thank the reviewer for raising this important point, it is well taken. We re-
calculated the correlations between differential species and the patients' clinical metadata
by using R package ppcor (version 1.1). Specifically, we assessed the correlation between
individual clinical features and species, with other clinical metadata treated as covariates.
The results of these analyses are shown in Figure 1f of the revised version of our
manuscript and consistent with our previous analysis. In addition to our previous findings
we also observed a significantly positive association between *Klebsiella pneumoniae* and
UAS7 and a significantly negative association between *Roseburia hominis* and LPS
concentration.

(2) Additionally, the study lacks information on the adj-p values for microbiota and

metabolomic analysis. It would be interesting to see how many of the presented data
 remain significant after correcting for multiple comparisons using p.adjust. While the
 study's message may not change, and the most critical taxa will likely remain significant,
 the authors should correct all the statistics presented in their manuscript. I will suggest
 using a pipeline such as MaAsLin2, CoDaSeq or ALDEx2.

**Response:** We thank the reviewer for this helpful suggestion. We agree and we used R
 package MaAsLin2 (version 1.6.0) to re-identify differential species in CSU, with BH
 correction and maximum significance set at 0.20. As shown in Figure 1b of the revised
 version of our revised manuscript, compared to our previous results, the most critical taxa
 remained significant. Specifically, 12 species were consistently different including
 *Ruminococcus gnavus*, *Bacteroides stercoris*, *Klebsiella pneumonia*, *Escherichia coli*,
 *Ruminococcus obeum*, *Roseburia hominis*, *Bacteroides cellulosilyticus*, *Odoribacter*
 *splanchnicus*, *Alistipes shahii*, *Alistipes putredinis*, *Alistipes onderdonkii*, and *Coprococcus*
 *catus*. In addition, we identified three new differential species, i.e. *Veillonella parvula*,
 *Bilophila wadsworthia*, and *Bacteroides intestinalis*.

Fig.1b

 (3) The authors classified the bacteria into two categories: SCFA producers and
 pathobionts, which appear to be accurate based on the taxa identified in their study.
 However, it would be beneficial if they further clarified how they categorized these
 bacteria.

**Response:** Thank you for your helpful suggestion. We now provide further clarification
 and information on the categorization and characteristics of 15 different bacteria. This
 information is summarized in the Supplemental Table 2 of the revised version of our
 manuscript.

Supplementary Table 2 The bacteria alteration between HC and CSU group

Species	Opportunistic pathogens	SCFA producers	Harmful effect on intestinal permeability	Reference
Odoribacter splanchnicus	No	Yes	No	Ref: 2,3
Ruminococcus obeum	No	Yes	No	Ref: 4-6
Alistipes putredinis	Yes	Yes	No	Ref: 7,8
Alistipes shahii	Yes	Yes	No	Ref: 9,10
Alistipes onderdonkii	Yes	Yes	No	Ref: 8,11
Bacteroides cellulosilyticus	Yes	Yes	No	Ref: 12,13
Coprococcus catus	No	Yes	No	Ref: 5,14,15
Roseburia hominis	No	Yes	No	Ref: 16,17
Escherichia coli	Yes	No	Yes	Ref: 18,19
Klebsiella pneumoniae	Yes	No	Yes	Ref: 19,20
Bacteroides stercoris	Yes	Yes	No	Ref: 21,22
Ruminococcus gnavus	Yes	No	Yes	Ref: 23,24
Bilophila wadsworthia	Yes	No	Yes	Ref: 1,25
Veillonella parvula	Yes	No	Yes	Ref: 26,27
Bacteroides intestinalis	Yes	Yes	No	Ref: 28

(4) What are the keystone taxa in each network analysis performed? Does "the key hub
 species" mean keystone taxa? If so, can the authors calculate the Between Centrality Score to
 identify the most influential taxa in each group?

**Response:** Thank you for your question. "The key hub species", indeed, means keystone taxa.

In this study, the networks were constructed and analyzed by using R package NetCoMi. Here,
 the hub species of each group (i.e. HC, CSU) were identified by "eigenvector" by default.

As suggested, we calculated the hub species of each group by "betweenness" and
 "eigenvector" , and the results are presented in Table 3 as follows:

Table 3: The calculated centrality values to compare network properties.

	HC	CSU	abs. diff.	adj.p-value
Betweenness centrality (normalized):				
s_Clostridium_hathewayi	0.045	0.435	0.389	0.009 **
s_Streptococcus_parasanguinis	0.256	0.019	0.237	0.063
s_Clostridium_bolteae	0.21	0.445	0.235	0.191
s_Bacteroides_intestinalis	0	0.228	0.228	0.059
s_Klebsiella_pneumoniae	0	0.191	0.191	0.042 *
s_Lachnospiraceae_bacterium_5_1_63FAA	0.179	0	0.179	0.082
s_Alistipes_shahii	0	0.148	0.148	0.384
s_Bacteroidales_bacterium_ph8	0.052	0.194	0.142	0.159
s_Bacteroides_fragilis	0.139	0	0.139	0.061
s_Ruminococcus_gnavus	0.059	0.197	0.138	0.413
Eigenvector centrality (normalized):				
s_Ruminococcus_obeum	1	0.002	0.998	0.016 *
s_Veillonella_parvula	0.011	0.974	0.963	0.025 *
s_Dorea_longicatena	0.912	0.001	0.912	0.036 *
s_Klebsiella_pneumoniae	0.027	0.924	0.898	0.002 **
s_Streptococcus_salivarius	0.04	0.92	0.88	0.099
s_Lachnospiraceae_bacterium_5_1_63FAA	0.839	0	0.839	0.03 *
s_Haemophilus_parainfluenzae	0.047	0.852	0.805	0.063
s_Streptococcus_parasanguinis	0.2	1	0.8	0.163
s_Dorea_formicigenerans	0.789	0.002	0.787	0.128
s_Eubacterium_hallii	0.629	0.001	0.628	0.225

Significance codes: ***: 0.001, **: 0.01, *: 0.05, .: 0.1

(5) Figure 2D and E: what are the level in control (non-treatment at all) or just antibiotic-
treated but no FMT performed groups of mice? It will be important to see the basal level.

**Response:** Thank you for your question. To answer it, we repeated the FMT of gut
microbiota from CSU patients and assessed recipient mice of the control group (no
treatment at all) and just antibiotic-treated. The results are shown in Figure 2a-d and h-
k of the revised version of the manuscript. Briefly, mice treated with antibiotics only but
not FMT had intestinal permeability similar to that in mice transplanted with healthy
human intestinal flora. Meanwhile, mice treated with antibiotics but not transplanted with
healthy human intestinal flora exhibited similar skin inflammation in PCA as compared to
mice transplanted with healthy human intestinal flora.

Fig. 2:

(6) Figure 3: Could the authors provide more information on the composition of the
 control group in their study? It is stated that a control group is an antibiotic-treated group,
 but it is unclear if there was a group of totally normal colonized SPF mice used as a control.
 Including such a control group would provide valuable information on the effects of
 antibiotics on the phenotypes observed in the study, and whether they can be reversed
 to a basal state. Therefore, it would be beneficial if the authors could provide clarification

on whether a group of completely untreated SPF mice was used as a control and, if not,
the rationale for selecting an antibiotic-treated group as the control.

**Response:** We thank the reviewer for this helpful suggestion. We repeated the *Roseburia*
*hominis* monobacterial transplantation experiment, where mice are subjected to
antibiotic treatment (Abx), transfer of *Roseburia hominis* (RHO), and passive cutaneous
anaphylaxis (PCA). In addition to the three groups in our original experiment, i.e. the Abx
group, the Abx+PCA group, and the Abx+RHO+PCA group, we included a group of mice
that did not receive antibiotic treatment before PCA (Non-Abx+PCA) and a group of
mice that did not receive antibiotic treatment before RHO transfer and PCA (Non-
Abx+RHO+PCA).

Skin inflammatory responses induced by PCA were slightly more severe in the
Abx+PCA group as compared to non-Abx+PCA mice, but the difference was not
statistically significant. Abx and non-Abx mice transplanted with RHO showed similar
elevated abundance of RHO and reduced PCA responses. Taken together, these
additional experiments confirm our previous findings and extend them, by showing that
the protection from skin inflammatory responses by transferred RHO does not require
prior antibiotic treatment. The results of these additional experiments are shown in Figure
3d-i of the revised version of our manuscript.

Fig. 3

(7) While reviewing the study, I attempted to access the xlsx files containing the raw data
 and statistical results provided by the authors. However, I found the files to be poorly
 labeled, with names such as "420201_0_extended_data_4098726_ry2c9.xlsx" or
 "420201_0_source_data_4098731_rr62c9.xlsx". Although the authors' effort to provide all
 the raw data is appreciated, it was time-consuming to navigate between the manuscript
 tracking system and the data files to identify the correct table number. While this is a
 minor issue, I would suggest that the authors deposit their files with more informative
 annotations in the future to make them more accessible to readers.

**Response:** Thank you for pointing this out, we agree that it should be easy for readers to
 understand and work with the raw data provided. We edited the excel sheets accordingly.

(8) Furthermore, a more significant concern is the absence of legends for the tables. I was

unable to locate the legends for the tables, and in some cases, some xlsx files contained
multiple tabs or several panels within the same tab. While this may be the format required
by the journal, I found it to be an unusual way of reporting data. Providing clear legends
and organizing the data in a more intuitive manner would improve the readability and
usability of the raw data for readers.

**Response:** The reviewer is right and we added clear legends to all tables.

Questions for the methodology section:

1 - Can you provide more information about the collection tubes that contain DNA
preservation solution? The methodology section does not specify the exact type or brand
of tubes used. It would be helpful to know the specific details of the tubes to better
understand the preservation method and potential effects on the microbiota profiles.

**Response:** Thanks for your question. The tubes we used are from Tinygene Biologicals,
Shanghai, China, Product No. GWF01-A. These stool preservation tubes are sterile
products, which minimizes contamination of samples by microorganisms and microbial
residual DNA. The stool preservation tubes contain the stool preservation fluid and have
a sterile spoon attached to the lid. We take samples by unscrewing the sterile tube, using
the spoon to take a sample and immediately putting it back into the tube and tightening
the lid, i.e. the process of taking samples is strictly aseptic.

2 - Have you ever sequenced the cotton swab itself to determine if it contributes to the
microbiota profile? Despite the authors' efforts to exclude extraneous DNA, it is important
to consider potential contamination from the swab. It would be interesting to know if the
swab has its own distinct microbial signature and to what extent it may affect the results.

**Response:** Thank you for your interesting question. The stool collection was done with a
sterile spoon, rather than a cotton swab.

3 - Line 400: 100um screen. What is this exactly?

**Response:** Thank you for your question. We filtered the participants' fecal suspensions

with a 100um sterile screen, which is a nylon screen with a uniform mesh size of 100 µm.
This is done to filter out messy material such as food residue from the feces.

4 - Macrogenomic sequencing. I am not aware of this terminology. Would it be more
appropriate to refer to this as amplicon sequencing or another more common term?

**Response:** Thanks for your careful review and catching this error. This is meant to say
metagenomic sequencing, and we corrected this in the revised version of our manuscript.

5 - The methodology section states that the authors sequenced the v3-v4 region using
amplicon sequencing, but then fragmented the amplicon. Could you clarify the details of
the primers used and the size of the amplicon? If the amplicon was already shorter than
600bp, it would be helpful to explain why it was necessary to fragment it further.

**Response:** Thanks for your question and suggestion for clarification. The primer
sequences we used were: 341F: CCTAYGGGRBGCASCAG; 806R:
GGACTACNNGGTATCTAAT. The length of the amplified region was 465bp. We
interrupted DNA before performing V3-V4 amplification, not after v3-v4 amplification.
The primer sequences we used were: 341F: CCTAYGGGRBGCASCAG; 806R:
GGACTACNNGGTATCTAAT. the length of the amplified region was 465bp. Our original
wording was flawed, and we corrected this in the revised version of our manuscript as
follows: "Genomic DNA was cut into fragments with an average of ~400bp using Covaris
M220 (Gene Company Limited, China). Then, the v3-v4 region of the 16S rRNA gene was
amplified with specific primers with connector markers, and paired-end sequencing was
done using the Illumina NovaSeq6000 platform and NovaSeq Reagent Kits (Illumina Co.,
Ltd., USA)."

Line 132: Supplementary Figure 1. I couldn't find this Figure.

**Response:** Thanks for your detailed review. Supplementary Table 1 was mistakenly
labelled "Supplementary Figure 1". We have corrected the mistake in the revised version
of our manuscript.

Line 146: I believe that the Figure referred to in this paragraph is incorrectly labeled as
Extended Data Figure 4a.

**Response:** Thank you for catching this error. Figure 1d and the Extended Data Figure 4a
are duplicates, and we removed the Extended Data Figure 4a in the revised version of our
manuscript.

Line 176 - 181: This part of the study shows the longitudinal analysis however, it is a bit
confusing since it is not well described the terms in the text or legend. What do you mean
by status: high risk or low risk? Can you expand this results' interpretation so that we can
understand better? What do you mean by survival probability? Does anyone die from this
disease? Sorry for my ignorance.

**Response:** Thank for your helpful questions. In the revised version of our manuscript, we
improved the description of this analysis and its results. What we did was assess patients,
for five years, for CSU remission and recurrence, defined as the reappearance of CSU
signs and symptoms after complete remission and cessation of controller therapy. What
we found was that 17 of 22 CSU patients showed complete (spontaneous) remission and
7 of them experienced relapse, which was linked to low levels of SCFA-producing and
high levels of opportunistic pathogenic bacteria and high LPS blood levels as well as 15
differential species.

**References:**

Buhner, S., et al.

2004 Pseudoallergic reactions in chronic urticaria are associated with altered
gastroduodenal permeability. *Allergy* 59(10):1118-23.

Chen, C., et al.

2022 Gut microbiota regulate Alzheimer's disease pathologies and cognitive
disorders via PUFA-associated neuroinflammation. *71(11):2233-2252.*

Crestani, E., et al.

2020 Untargeted metabolomic profiling identifies disease-specific signatures in
food allergy and asthma. *J Allergy Clin Immunol* 145(3):897-906.

Dawson, P. A., et al.

2009 Reduced mucin sulfonation and impaired intestinal barrier function in the
hyposulfataemic NaS1 null mouse. *Gut* 58(7):910-9.

Doreswamy, V., and D. B. Peden

2011 Modulation of asthma by endotoxin. *Clin Exp Allergy* 41(1):9-19.

Fang, Z., et al.

2022 Bifidobacterium longum mediated tryptophan metabolism to improve
atopic dermatitis via the gut-skin axis. *Gut Microbes* 14(1):2044723.

Folkerts, J., et al.

2020 Butyrate inhibits human mast cell activation via epigenetic regulation of
FcεRI-mediated signaling. 75(8):1966-1978.

Kumar, S., and A. Adhikari

2017 Dose-dependent immunomodulating effects of endotoxin in allergic airway
inflammation. *Innate Immun* 23(3):249-257.

Lamas, B., et al.

2016 CARD9 impacts colitis by altering gut microbiota metabolism of tryptophan
into aryl hydrocarbon receptor ligands. 22(6):598-605.

Lamichhane, S., et al.

2022 Dysregulation of secondary bile acid metabolism precedes islet
autoimmunity and type 1 diabetes. *Cell Rep Med* 3(10):100762.

Pijls, K. E., et al.

2014 Large intestine permeability is increased in patients with compensated liver
cirrhosis. *Am J Physiol Gastrointest Liver Physiol* 306(2):G147-53.

Sorribas, M., et al.

2019 FXR modulates the gut-vascular barrier by regulating the entry sites for
bacterial translocation in experimental cirrhosis. *J Hepatol* 71(6):1126-1140.

Vatanen, T., et al.

2016 Variation in Microbiome LPS Immunogenicity Contributes to Autoimmunity
in Humans. *Cell* 165(4):842-53.

Wang, X., et al.

2020 Aberrant gut microbiota alters host metabolome and impacts renal failure
in humans and rodents. *69(12):2131-2142.*

Woting, A., and M. Blaut

2018 Small Intestinal Permeability and Gut-Transit Time Determined with Low
and High Molecular Weight Fluorescein Isothiocyanate-Dextrans in C3H Mice.
*Nutrients* 10(6).

Zhang, Y., et al.

2022 Bacteroides species differentially modulate depression-like behavior via
gut-brain metabolic signaling. *Brain Behav Immun* 102:11-22.

Zhou, Y., et al.

2013 Aryl hydrocarbon receptor controls murine mast cell homeostasis. *Blood*
121(16):3195-204.

REVIEWERS' COMMENTS

Reviewer #1 (Remarks to the Author):

The authors have made a strong effort to respond to reviewer comments and I have no further scientific suggestions.

I do suggest to change 'mice and men' line 332, page 14 to 'mice and humans'.

Reviewer #2 (Remarks to the Author):

The authors adequately addressed most of my concerns. I believe the manuscript is greatly improved.

Reviewer #3 (Remarks to the Author):

I have checked the comments and feedback provided by the authors and find their responses to be satisfactory in addressing our queries. Pertaining to the comments from reviewer #1, I deem their responses to be sufficient and have no further questions on the matter.

Point-by-point response to the Reviewers

Date: 23-Aug-2023

Manuscript Number: NCOMMS-23-12292-T

Title of Article: Gut microbiota facilitate chronic spontaneous urticaria

Names of the Corresponding Authors: Jie Li, Marcus Maurer

Email Addresses of the Corresponding Authors: xyljie@csu.edu.cn,
marcus.maurer@charite.de

Dear reviewers,

Thank you for your positive feedback and comments on our manuscript “Gut microbiota facilitate chronic spontaneous urticaria”. And thank you for taking time again to review and improve our manuscript. We made corrections in response to your comments as detailed below and highlighted in yellow in the revised version of our manuscript.

REVIEWER COMMENTS

Reviewer #1 (Remarks to the Author):

The authors have made a strong effort to respond to reviewer comments and I have no further scientific suggestions.

I do suggest to change 'mice and men' line 332, page 14 to 'mice and humans'.

Response: We sincerely thank the reviewer for detailed review which help us to improve our manuscript. We changed the word 'mice and men' to 'mice and humans' in the new manuscript in page 13, line 322..

Reviewer #2 (Remarks to the Author):

The authors adequately addressed most of my concerns. I believe the manuscript is greatly improved.

Response: We sincerely thank the reviewer for carefully reading our article and making constructive and valuable comments on our article, which made our research more completely.

Reviewer #3 (Remarks to the Author):

I have checked the comments and feedback provided by the authors and find their responses to be satisfactory in addressing our queries. Pertaining to the comments from reviewer #1, I deem their responses to be sufficient and

have no further questions on the matter.

Response: We are deeply grateful to the reviewer for his/her careful reading and examination of our research and for the suggestions that are essential to improve our work.